# Approximate Bayesian Computation with Path Signatures

Joel Dyer[1,2]                Patrick Cannon[3]                Sebastian M. Schmon[3]

[1]Department of Computer Science, University of Oxford
[2]Institute for New Economic Thinking, University of Oxford
[3]No affiliation

## Abstract

Simulation models often lack tractable likelihood functions, making likelihood-free inference methods indispensable. Approximate Bayesian computation generates likelihood-free posterior samples by comparing simulated and observed data through some distance measure, but existing approaches are often poorly suited to time series simulators, for example due to an independent and identically distributed data assumption. In this paper, we propose to use path signatures in approximate Bayesian computation to handle the sequential nature of time series. We provide theoretical guarantees on the resultant posteriors and demonstrate competitive Bayesian parameter inference for simulators generating univariate, multivariate, and irregularly spaced sequences of non-*iid* data.

## 1 INTRODUCTION

Simulation models are an increasingly popular tool in a broad range of scientific disciplines including cosmology [Alsing et al., 2018], economics [Geanakoplos et al., 2012], and the biological sciences [Christensen et al., 2015]. A drawback of such models is that, while they are straightforward to sample from, their complexity typically does not allow for explicit evaluation of the associated likelihood function. Consequently, traditional approaches to statistical inference are infeasible and alternative likelihood-free inference (LFI) methods are usually adopted.

Many LFI approaches have been proposed. One of the most widely used LFI methods is approximate Bayesian computation (ABC) [Tavaré et al., 1997, Pritchard et al., 1999, Beaumont et al., 2002], in which the Bayesian posterior distribution is approximated by sampling parameters $\theta$ from a prior distribution and synthetic datasets $\mathbf{x}$ from a stochastic simulator – with likelihood denoted $p(\mathbf{x} \mid \boldsymbol{\theta})$ – and compar-

ing the output $\mathbf{x}$ with real data $\mathbf{y}$. If the simulator output is sufficiently 'close' to the observation, then $\boldsymbol{\theta}$ is retained as a sample from the approximate posterior distribution; otherwise, it is discarded.

However, measuring closeness between model outputs is known to be challenging. This is particularly the case for time series data, which can exhibit complex dependency structures and may be multivariate and sampled at irregular time intervals. A common approach is to attempt to distil important features of the data using summary statistics and compare these instead (see e.g. Prangle [2018]). In practice, informative summary statistics are difficult to craft, which presents a trade off: a poor choice can materially degrade ABC-based posterior approximations, yet constructing a sufficiently powerful choice can require substantial domain expertise, problem insight, and costly experimentation (see e.g. Drovandi and Frazier [2021] for a recent comparison of methods with and without summaries).

In other approaches the engineering of summary statistics is bypassed altogether in favour of distances on the full dataset [e.g. Park et al., 2016, Jiang, 2018, Bernton et al., 2019, Nguyen et al., 2020]. However, in many such cases the focus is on *iid* data, with non-*iid* or sequential data appearing as an afterthought. The result of this is that there is a scarcity of automatic approaches to performing approximate Bayesian inference for generic dynamic, stochastic simulation models in the ABC literature. Developing automatic approaches to ABC that are more tailored to simulators generating sequences of dependent points will thus increase the ease with which ABC methods can be deployed in a broader range of real-world inference settings.

In response to this challenge, we present here two novel methods for performing ABC for time series models that bypass the difficult problem of manually constructing summary statistics for sequential data. Our approach leverages so-called *path signatures*, a key object in the mathematics of rough path theory and the theory of controlled differential equations [see e.g. Lyons et al., 2007, Lyons, 2014],

to generate ABC schemes that places sequential data at centre stage. Signatures have been employed successfully in a variety of machine learning tasks (see, e.g., [Li et al., 2017, Moore et al., 2019]), and constitute a natural feature set for multivariate and even irregularly sampled sequential data [Salvi et al., 2021]. We demonstrate that the path signature can be employed in two different ways to construct useful distance measures for time series data in ABC: either directly as a summary statistic, or in the context of a regression-based semi-automatic ABC approach. We further show that such approaches can recover more accurate posteriors than existing techniques.

## 2 BACKGROUND

In this section, we recapitulate some standard approaches to ABC with an emphasis on time series data, and provide an overview of path signatures. Appendix B expands on this introduction to path signatures for the unfamiliar reader.

### 2.1 APPROXIMATE BAYESIAN COMPUTATION

Let $\mathcal{X}^n$ be the space of length $n$ sequences taking values in a set $\mathcal{X}$. Suppose we have time series data $\mathbf{y} = (\mathbf{y}_{t_1}, \mathbf{y}_{t_2}, \ldots, \mathbf{y}_{t_n})$ with each $\mathbf{y}_{t_i} \in \mathcal{X}$, observed at real times $0 = t_1 < t_2 < \ldots < t_n = T$, and assumed to have been drawn from a model with measure $\mu_{\boldsymbol{\theta}}$ parameterised by $\boldsymbol{\theta} = (\boldsymbol{\theta}_1, \ldots, \boldsymbol{\theta}_p) \in \boldsymbol{\Theta} \subseteq \mathbb{R}^p$. We assume that $\mu_{\boldsymbol{\theta}}$ has density $p_{\boldsymbol{\theta}}$ with respect to the Lebesgue measure. Given a prior density $\pi$ (also wrt Lebesgue) on $\boldsymbol{\Theta}$, the central object in Bayesian inference is the posterior distribution

$$\pi(\boldsymbol{\theta} \mid \mathbf{y}) \propto p_{\boldsymbol{\theta}}(\mathbf{y})\pi(\boldsymbol{\theta}). \quad (1)$$

For simulation models, evaluating the likelihood function $p_{\boldsymbol{\theta}}(\mathbf{y})$ is commonly intractable, making standard Bayesian approaches to posterior inference such as Markov chain Monte Carlo (MCMC) infeasible.

In such scenarios, an established alternative is approximate Bayesian computation (ABC) [Tavaré et al., 1997, Pritchard et al., 1999, Beaumont et al., 2002] which allows the user to approximate the true posterior (1) using only forward simulations. Broadly, the user specifies summary statistics $\mathbf{s} : \mathcal{X}^n \to \mathcal{S}$ (usually $\mathcal{S} = \mathbb{R}^k$ for some $k \geq 1$), and a distance measure $\rho$; $p_{\boldsymbol{\theta}}(\mathbf{y})$ is then approximated as

$$\tilde{p}_\varepsilon\{\mathbf{s}(\mathbf{y}) \mid \boldsymbol{\theta}\} = \int K_\varepsilon \left[\rho\{\mathbf{s}(\mathbf{y}), \mathbf{s}(\mathbf{x})\}\right] p_{\boldsymbol{\theta}}(\mathbf{x}) \, \mathrm{d}\mathbf{x}, \quad (2)$$

where $K_\varepsilon(\cdot) = K(\cdot/\varepsilon)/\varepsilon$ is a kernel function with bandwidth parameter $\varepsilon$. The resulting ABC posterior is then

$$\pi_\varepsilon\{\boldsymbol{\theta} \mid \mathbf{s}(\mathbf{y})\} \propto \tilde{p}_\varepsilon\{\mathbf{s}(\mathbf{y}) \mid \boldsymbol{\theta}\}\pi(\boldsymbol{\theta}). \quad (3)$$

The approach as presented above leaves open a plethora of possible choices for $\mathbf{s}$, $\rho$ and $K_\varepsilon(\cdot)$. We summarise here some of the most common and well-known choices.

**Rejection ABC**  The standard rejection ABC (REJ-ABC) algorithm corresponds to choosing a uniform kernel $K_\varepsilon(\cdot) \propto \mathbb{1} \left(\cdot \leq \varepsilon\right)$. The choice of threshold $\varepsilon$ is left to the experimenter, and for example may be determined in advance of the inference procedure, or chosen after simulation time such that a certain proportion of the total simulation budget is retained [Cornuet et al., 2008].

**Semi-automatic ABC**  Fearnhead and Prangle [2012] propose semi-automatic ABC (SA-ABC), in which an estimate of the posterior mean, $\mathbf{s}(\mathbf{y}) = \mathbb{E}(\boldsymbol{\theta} \mid \mathbf{y})$, acts as the summary statistic, and the Euclidean distance is used as $\rho$. Given a set of $N$ training data points $(\mathbf{x}^{(i)}, \boldsymbol{\theta}^{(i)}) \sim p_{\boldsymbol{\theta}}(\mathbf{x})\pi(\boldsymbol{\theta})$, $i = 1, \ldots, N$, and a candidate vector $\mathbf{g}(\cdot)$ of $J$ summary statistics, the method performs vector-valued regression from $\mathbf{g}(\mathbf{x}^{(i)})$ to $\boldsymbol{\theta}^{(i)}$ to estimate $\mathbf{s}(\mathbf{y})$. A drawback of this method is that it requires the construction of an initial set of candidate summaries, which would need to be informative. Other approaches in this vein include Nakagome et al. [2013], in which the authors propose the use of SA-ABC using kernel ridge regression to exploit the nonlinearities induced by kernel methods.

**K2-ABC**  Park et al. [2016] propose double kernel ABC (K2-ABC), which bypasses the problem of constructing summary statistics for *iid* data by using the maximum mean discrepancy (MMD) between (a) the simulator's distribution $f_{\boldsymbol{\theta}}$, where $\mathbf{x} = (\mathbf{x}_1, \ldots, \mathbf{x}_n) \sim p_{\boldsymbol{\theta}}(\mathbf{x}) = \prod_{i=1}^n f_{\boldsymbol{\theta}}(\mathbf{x}_i)$, and (b) the true density $f^*$ giving rise to the *iid* observations comprising $\mathbf{y}$, respectively. That is, from a kernel $\kappa : \mathcal{X} \times \mathcal{X} \to \mathbb{R}$, the discrepancy between $\mathbf{x}$ and $\mathbf{y}$ is taken to be

$$\text{MMD}^2 = \|\mathbb{E}_{\mathbf{z} \sim f_{\boldsymbol{\theta}}}[\kappa(\mathbf{z}, \cdot)] - \mathbb{E}_{\mathbf{z}' \sim f^*}[\kappa(\mathbf{z}', \cdot)]\|_{\mathcal{H}}^2, \quad (4)$$

where $\mathcal{H}$ is the reproducing kernel Hilbert space (RKHS) associated with $\kappa$. In this way, the choice of summary statistics (e.g. as required in SA-ABC) can be seen as being replaced by the choice of kernel $\kappa$. For time series data, the authors suggest that the dependency structure can be ignored, and that the observation $\{\mathbf{y}_i : i = 1, \ldots, n\}$ and simulation output $\{\mathbf{x}_i : i = 1, \ldots, m\}$ can still be treated as *iid* data from the marginal densities $f_{\boldsymbol{\theta}}$ and $f^*$, respectively.

**Wasserstein ABC (W-ABC)**  Bernton et al. [2019] propose to use as its measure of discrepancy the $p$-Wasserstein distance between the empirical distribution of observations $\mathbf{y} = (\mathbf{y}_1, \mathbf{y}_2, \ldots, \mathbf{y}_n)$, and simulated data $\mathbf{x} = (\mathbf{x}_1, \mathbf{x}_2, \ldots, \mathbf{x}_m)$, with $\mathbf{y}_i, \mathbf{x}_j \in \mathbb{R}^d$. That is, the distance $\rho$ is taken to be

$$\mathcal{W}_p(\mathbf{y}, \mathbf{x})^p = \inf_{\gamma \in \Gamma_{n,m}} \sum_{i=1}^n \sum_{j=1}^m \rho_0(\mathbf{y}_i, \mathbf{x}_j)^p \gamma_{ij} \quad (5)$$

where $\rho_0$ is a distance on $\mathbb{R}^d$ and $\Gamma_{n,m}$ is the set of $n \times m$ matrices with non-negative entries, columns summing to $m^{-1}$, and rows summing to $n^{-1}$. The authors discuss

multiple strategies to account for the structured and ordered nature of time series data, such as the *Wasserstein curve matching* distance, in which a time augmentation $\mathbf{y}_{t_i} \mapsto (t_i, \mathbf{y}_{t_i})$ is applied to the data, and the following ground distance between elements of the sequence used:

$$\rho_0\{(t_i, \mathbf{y}_{t_i}), (t_j, \mathbf{x}_{t_j}); \lambda\} = \|\mathbf{y}_{t_i} - \mathbf{x}_{t_j}\| + \lambda|t_i - t_j|. \quad (6)$$

In the above, $\lambda > 0$ is a free parameter that interpolates the distance in (5) between the sum of Euclidean distances $\sum_i \|\mathbf{y}_{t_i} - \mathbf{x}_{t_i}\|$ (when $n = m$) and the Wasserstein distance between the empirical marginal distributions of $\mathbf{y}$ and $\mathbf{x}$. Such an approach is however of limited suitability for time series data: the curve matching distance will not in general respect the ordering of the observations in $\mathbf{x}$ and $\mathbf{y}$, and will ultimately still permit permutations of their elements (see Appendix A for a simple example).

## 2.2 PATH SIGNATURES

Let $\mathcal{H}$ be a Hilbert space carrying an inner product $\langle \cdot, \cdot \rangle_{\mathcal{H}}$, and $h : [0, T] \to \mathcal{H}$ be a $\mathcal{H}$-valued path on the interval $[0, T]$. Further, let $\zeta(0, T) = \{t_1, \ldots, t_n\}$ denote a finite partition of the interval $[0, T]$, with $0 = t_1 < \cdots < t_n = T$. Throughout, we will consider $\mathcal{H}$-valued paths of bounded variation over the interval $[0, T]$, i.e. paths for which

$$\|h\|_{1-\text{var}} := \sup_{\zeta(0,T)} \sum_{i=1}^{n-1} \|h_{t_{i+1}} - h_{t_i}\|_{\mathcal{H}} < \infty,$$

where the supremum is taken over all finite partitions of the domain and $n = |\zeta(0, T)|$. We denote with $BV_{[0,T]}(\mathcal{H})$ the space of such paths. By defining the product Hilbert space

$$\prod_{m \geq 0} \mathcal{H}^{\otimes m} := \mathbb{R} \oplus \mathcal{H} \oplus (\mathcal{H} \otimes \mathcal{H}) \oplus \cdots \oplus \mathcal{H}^{\otimes m} \oplus \ldots, \quad (7)$$

endowed with an addition operation, inner product, and norm acting on any $A = (a_0, a_1, \ldots), B = (b_0, b_1, \ldots) \in \prod_{m \geq 0} \mathcal{H}^{\otimes m}$ as, respectively,

$$A + B := (a_0 + b_0, a_1 + b_1, \ldots), \quad (8)$$

$$\langle A, B \rangle := \sum_{m \geq 0} \langle a_m, b_m \rangle_{\mathcal{H}^{\otimes m}} \quad (9)$$

where

$$\langle u_1 \otimes \cdots \otimes u_m, v_1 \otimes \cdots \otimes v_m \rangle_{\mathcal{H}^{\otimes m}} = \prod_{j=1}^{m} \langle u_j, v_j \rangle_{\mathcal{H}},$$

and

$$\|A\| := \sqrt{\sum_{m \geq 0} \|a_m\|_{\mathcal{H}^{\otimes m}}^2}, \quad (10)$$

the *path signature* [see e.g. Lyons et al., 2007] of $h \in BV_{[0,T]}(\mathcal{H})$, denoted $\text{Sig}(h)$, maps $h$ to an infinite series of tensors as

$$h \mapsto \{1, S_1(h), S_2(h), \ldots\} \in E \subset \prod_{m \geq 0} \mathcal{H}^{\otimes m}. \quad (11)$$

In the above, $E$ is the subspace of $\prod_{m \geq 0} \mathcal{H}^{\otimes m}$ consisting of the elements of the product Hilbert space that have finite norm. The terms of the signature are defined as

$$S_m(h) := \int_0^T \mathrm{d}h^{\otimes m} = \underset{0 \leq t_1 < \cdots < t_m \leq T}{\int \cdots \int} \mathrm{d}h_{t_1} \otimes \cdots \otimes \mathrm{d}h_{t_m}.$$

We adopt the convention that $\mathcal{H}^{\otimes 0} = \mathbb{R}$. The number $m$ of integrals comprising the terms of the signature is often referred to as the *depth* of that signature term, and we adopt this terminology throughout. The collection of all such integrals at every depth $m \geq 0$ is a set of statistics for path-valued random variables that describe geometric features of the path and behave analogously to monomials.

The following simple example of a two-dimensional path provides a demonstration of the sort of geometric information captured by the signature:

**Example 1** (Example 2.3, Kiŕaly and Oberhauser [2019]). *Let $h_t$ take values in $\mathbb{R}^2$, $h_t = (a_t, b_t)$. Then $\mathrm{d}h_t = (\mathrm{d}a_t, \mathrm{d}b_t)$, such that*

$$S_1(h) = \begin{bmatrix} \int_{t=0}^{T} \mathrm{d}a_t \\ \int_{t=0}^{T} \mathrm{d}b_t \end{bmatrix}, \quad and$$

$$S_2(h) = \begin{bmatrix} \int_{t'=0}^{T} \int_{t=0}^{t'} \mathrm{d}a_t \mathrm{d}a_{t'} & \int_{t'=0}^{T} \int_{t=0}^{t'} \mathrm{d}a_t \mathrm{d}b_{t'} \\ \int_{t'=0}^{T} \int_{t=0}^{t'} \mathrm{d}b_t \mathrm{d}a_{t'} & \int_{t'=0}^{T} \int_{t=0}^{t'} \mathrm{d}b_t \mathrm{d}b_{t'} \end{bmatrix}.$$

*In the above, the variables $t$ and $t'$ are dummy time indices that are being integrated over. These terms can be further interpreted geometrically: the terms in $S_1(h)$ capture the increments along each dimension, while the off-diagonal elements of $S_2(h)$ capture the areas above and below the curve; see Figure 1. Higher order terms capture higher order notions of area that are harder to visualise and interpret.*

Signatures have a number of desirable properties; for example, they are a *universal nonlinearity*. This means that for any compact set $\mathcal{K}$ of paths of bounded variation, any continuous, real-valued function $f$ on $\mathcal{K}$ can be approximated uniformly by linear functionals of the signature, i.e. for any $\varepsilon > 0$ there exists a linear functional $L$

$$\sup_{h \in \mathcal{K}} \left| f(h) - L\left[\text{Sig}(h)\right] \right| < \varepsilon.$$

Appendix B.4 provides further details. Further, signatures have the desirable property of being an essentially *injective* map (see Appendix B.2), an important consequence of which in the context of approximate inference is that the path signature for a sequence of data can be seen as a *sufficient statistic*, since by the Fisher-Neyman factorization theorem [see, e.g., Schervish, 1995, Theorem 2.21] an injective function of a sufficient statistic is also sufficient.

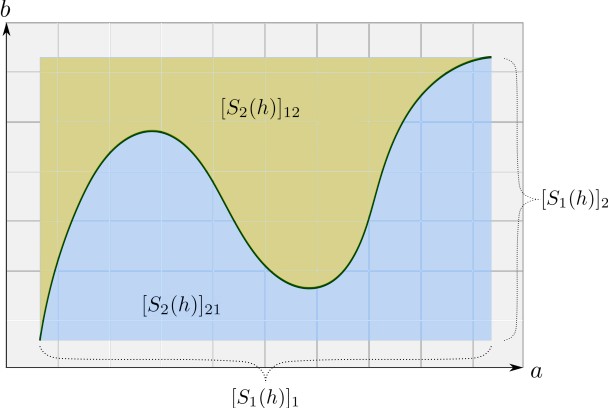

Figure 1: Geometric interpretation of the signature terms for the two-dimensional path from Example 1, shown as the dark green curve. Depth-1 terms correspond to the increments $a_T - a_0$ and $b_T - b_0$, while the depth-2 terms $[S_2(h)]_{21}$ and $[S_2(h)]_{12}$ correspond to the blue and yellow areas, respectively.

### 2.2.1 The Signature Kernel

The signature can be kernelised following Kiŕaly and Oberhauser [2019]:

**Definition 1** (Signature kernel, Kiŕaly and Oberhauser [2019]). *The signature kernel acts on $h, g \in BV_{[0,T]}(\mathcal{H})$ as*

$$k : (h, g) \mapsto \langle \mathrm{Sig}(h), \mathrm{Sig}(g) \rangle \in \mathbb{R}, \qquad (12)$$

*where the inner product is defined as in Equation* (9).

A key insight of Kiŕaly and Oberhauser [2019] was to recognise that evaluation of the signature kernel – which operates on *paths* in $\mathcal{H}$ – can be performed using only evaluations of an inner product $\kappa$ that operates on *points* in the path, amounting to a kernel trick for the signature kernel. Kiŕaly and Oberhauser [2019] further describe an efficient Horner scheme to evaluate a truncated signature kernel that approximates Equation (12). Salvi et al. [2021] extend this work by recognising that the signature kernel solves a Goursat partial different equation, permitting numerical estimation of the signature kernel using finite difference methods.

### 2.2.2 Path Signatures in Practice

In light of their interesting and useful properties described above, signatures can be seen as a canonical feature transformation for path-valued random variables. However, there exists an incongruity between our discussion so far and the scenarios faced in real-world settings: in reality and from the output of simulation models, we tend to observe discretely sampled data $\mathbf{x} = (\mathbf{x}_{t_1}, \mathbf{x}_{t_2}, \ldots, \mathbf{x}_{t_n})$ at times $0 = t_1 < t_2 < \cdots < t_n = T$, where $\mathbf{x}_t \in \mathcal{X}$ for

some finite-dimensional space $\mathcal{X}$ (for example $\mathbb{R}^d$ for some $d \geq 1$), rather than continuous paths $x \in BV_{[0,T]}(\mathcal{H})$. The incongruity is fixed in the following way:

(a) As noted by Kiŕaly and Oberhauser [2019], the aforementioned signature kernel trick can be used to introduce nonlinearities and embed the $\mathcal{X}$-valued sequence $\mathbf{x}$ in a Hilbert space. In particular, by choosing a reproducing kernel $\kappa : \mathcal{X} \times \mathcal{X} \to \mathbb{R}$ with RKHS $\mathcal{H}$ and canonical feature map $\kappa(\mathbf{x}_t, \cdot) \in \mathcal{H}$ as the inner product on the data space $\mathcal{X}$, we may implicitly construct a sequence $(\kappa(\mathbf{x}_{t_1}, \cdot), \kappa(\mathbf{x}_{t_2}, \cdot), \ldots, \kappa(\mathbf{x}_{t_n}, \cdot))$ of points in $\mathcal{H}$ from sequences of data in $\mathcal{X}$.

(b) To construct continuous paths from the discrete sequence above, an interpolation scheme is employed. While many interpolation schemes are possible, the most common is linear interpolation. Indeed, Kiŕaly and Oberhauser [2019] and Salvi et al. [2021] assume a linear interpolation to construct *discretised* signature kernels operating on sequences of points, and we use this interpolation scheme throughout this work.

By combining these two steps, we progress from a sequence $\mathbf{x}$ of points in $\mathcal{X}$ to a $\mathcal{H}$-valued, piecewise linear path $h$, which for $i = 1, \ldots, n-1$ and $t \in [t_i, t_{i+1}]$ is given by

$$h_t := \kappa(\mathbf{x}_{t_i}, \cdot) + \frac{t - t_i}{t_{i+1} - t_i} \{ \kappa(\mathbf{x}_{t_{i+1}}, \cdot) - \kappa(\mathbf{x}_{t_i}, \cdot) \}. \quad (13)$$

Piecewise linear paths constructed in this way are naturally of bounded variation if, for example, $\kappa$ is a continuous and/or uniformly bounded kernel (see Proposition 6 in Appendix C). We will assume this throughout, such that all observed sequences in $\mathcal{X}$ lift to piecewise linear paths of bounded variation in $\mathcal{H}$ under the feature map corresponding to $\kappa$.

## 3 METHODS

Given its unique properties, the path signature and its associated kernel are natural candidates for feature maps and discrepancy measures in ABC to handle time series data of different kinds. In this section, we introduce and investigate two techniques for incorporating signatures into ABC.

### 3.1 SIGNATURE ABC

The first approach we consider entails using the signature directly as a summary statistic in ABC. Though signatures are infinite-dimensional objects, we can leverage their kernel representation (see Definition 1) to compute the distance between two sequences $\mathbf{x}, \mathbf{y}$ as the norm induced by the associated signature inner product. That is, for two time series $\mathbf{x}$ and $\mathbf{y}$, we can interpret the signature of their lifted paths as a *sufficient* summary statistic, $\mathbf{s}(\mathbf{x}) = \mathrm{Sig}(\mathbf{x})$, and

compute

$$\rho\{\mathbf{s}(\mathbf{x}), \mathbf{s}(\mathbf{y})\} := \|\mathrm{Sig}(\mathbf{x}) - \mathrm{Sig}(\mathbf{y})\|^2 \qquad (14)$$
$$= k(\mathbf{x}, \mathbf{x}) + k(\mathbf{y}, \mathbf{y}) - 2\,k(\mathbf{x}, \mathbf{y}),$$

where $k(\mathbf{x}, \mathbf{y}) = \langle \mathrm{Sig}(\mathbf{x}), \mathrm{Sig}(\mathbf{y}) \rangle$. The resulting distance can be computed easily using existing software[1] and used to derive an ABC posterior via Equations (2)-(3). For example, it may be embedded in rejection ABC, yielding

$$\pi_\varepsilon(\boldsymbol{\theta} \mid \mathbf{y}) \propto \pi(\boldsymbol{\theta}) \int \mathbb{1}\left(\|\mathrm{Sig}(\mathbf{x}) - \mathrm{Sig}(\mathbf{y})\|^2 \leq \varepsilon\right) \mu_{\boldsymbol{\theta}}(\mathrm{d}\mathbf{x}),$$

as the ABC posterior. We term this approach Signature ABC (S-ABC). Injectivity of the path signature, and continuity of the norm the inner product induces, further guarantees the asymptotic correctness of this S-ABC posterior as $\varepsilon \to 0$ with $n$ fixed:

**Proposition 1.** *Let* $\mathcal{X} := \mathbb{R}^d$, $\mathbf{y} = (\mathbf{y}_1, \ldots, \mathbf{y}_n) \in \mathcal{X}^n$, *and let* $\rho$ *be as in Equation* (14)*, resulting in the S-ABC posterior* $\pi_\varepsilon$*. Suppose the density function* $p_{\boldsymbol{\theta}}(\mathbf{x})$ *satisfies*

$$\sup_{\boldsymbol{\theta} \in \Theta \setminus \mathcal{N}_\Theta} p_{\boldsymbol{\theta}}(\mathbf{y}) < \infty,$$

*where* $\mathcal{N}_\Theta$ *is a set such that* $\pi(\boldsymbol{\theta}) = 0 \, \forall \boldsymbol{\theta} \in \mathcal{N}_\Theta$*, and that there exists* $\bar{\varepsilon} > 0$ *such that*

$$\sup_{\boldsymbol{\theta} \in \Theta \setminus \mathcal{N}_\Theta} \sup_{\mathbf{z} \in \mathcal{A}^{\bar{\varepsilon}}} p_{\boldsymbol{\theta}}(\mathbf{z}) < \infty,$$

*where* $\mathcal{A}^{\bar{\varepsilon}} := \{\mathbf{z} \in \mathcal{X}^n : \rho\{\mathbf{s}(\mathbf{y}), \mathbf{s}(\mathbf{z})\} \leq \bar{\varepsilon}\}$*. Then for any measurable* $\mathbf{B} \subset \Theta$*,*

$$\lim_{\varepsilon \to 0} \int_{\mathbf{B}} \pi_\varepsilon(\boldsymbol{\theta} \mid \mathbf{y}) \, \mathrm{d}\boldsymbol{\theta} = \int_{\mathbf{B}} \pi(\boldsymbol{\theta} \mid \mathbf{y}) \, \mathrm{d}\boldsymbol{\theta}. \qquad (15)$$

The proof is provided in Appendix C.

### 3.2 SIGNATURE REGRESSION ABC

We consider a second use of path signatures in ABC, namely in the SA-ABC method described by Fearnhead and Prangle [2012]. Given its status as a universal nonlinearity as discussed in Section 2.2, the path signature provides a natural basis for learning functions on sequences, and a natural set of summary statistics for the regression task required in SA-ABC. Regression on the full path signature is of course impossible, since the signature is infinite-dimensional. However, this may once again be circumvented using the signature kernel and corresponding kernel trick (see Definition 1) in kernel ridge regression [Hastie et al., 2001] to implicitly regress parameters onto the *full* signature, which is in a sense equivalent to using the infinitely long path signature as the candidate set of summary statistics

---

[1]E.g., `sigkernel` or `KSig`.

---

**Algorithm 1:** Rejection sampling scheme

**Input:** prior $\pi$, observation $\mathbf{y}$, distance function $\mathcal{D}(\cdot, \cdot)$, number of particles $N$, final sample size $M < N$;

**Result:** Posterior samples $\{\boldsymbol{\theta}^{(i)}\}_{i=1}^M$

**for** $i = 1, \ldots, N$ **do**
  Sample $\boldsymbol{\theta}^{(i)} \sim \pi$;
  Simulate $\mathbf{x}^{(i)} \sim p_{\boldsymbol{\theta}^{(i)}}$;
  Evaluate distance $\mathcal{D}(\mathbf{x}^{(i)}, \mathbf{y})$;
**end**

Retain the $M$ particles $\{\boldsymbol{\theta}^{(i)}\}_{i=1}^M$ with the lowest distances

---

in semi-automatic ABC. That is, using training examples $(\mathbf{x}^{(i)}, \boldsymbol{\theta}^{(i)}) \sim p_{\boldsymbol{\theta}}(\mathbf{x}) \, \pi(\boldsymbol{\theta}), i = 1, \ldots, R$, we find a function $\hat{\boldsymbol{\theta}}_j$ in the RKHS associated with the signature kernel $k$, which by the Representer Theorem has the following form for each component $\boldsymbol{\theta}_j, j = 1, \ldots, p$ of the $p$-dimensional parameters $\{\boldsymbol{\theta}^{(i)}\}_{i=1}^R$:

$$\hat{\boldsymbol{\theta}}_j(\mathbf{x}) = \sum_{i=1}^R \boldsymbol{\omega}_i^{(j)} k(\mathbf{x}, \mathbf{x}^{(i)}) \qquad (16)$$

with $\boldsymbol{\omega}^{(j)} = (G + \alpha I_R)^{-1} \boldsymbol{\psi}^{(j)}$, $\boldsymbol{\psi}^{(j)} = \left[\boldsymbol{\theta}_j^{(1)}, \boldsymbol{\theta}_j^{(2)}, \ldots, \boldsymbol{\theta}_j^{(R)}\right]'$ with $'$ denoting matrix transposition, $G_{mn} = k(\mathbf{x}^{(m)}, \mathbf{x}^{(n)})$, $I_R$ an $R \times R$ identity matrix, and $\alpha \geq 0$ is a regularisation parameter to be tuned. In this way, path signatures also enable the semi-automatic construction of summary statistics in SA-ABC. This approach to ABC is somewhat similar to that of Nakagome et al. [2013], who employ kernel ridge regression with a Gaussian RBF kernel to perform SA-ABC. Our approach differs, however, in that Nakagome et al. [2013] propose the use of hand-crafted summary statistics as input to the kernel ridge regression model, while we use the full data.

Once the data is summarised with this regression model, the discrepancy between simulation and observation is taken as the Euclidean distance between their corresponding outputs from the kernel ridge regression model. We herein refer to this approach as Signature regression ABC (SR-ABC). Further technical details are provided in Appendix D.1.

## 4 EXPERIMENTS

In this section, we present experiments comparing the performance of our methods, S-ABC and SR-ABC, against the use of the Wasserstein distance [Bernton et al., 2019] (W-ABC) and the MMD [Park et al., 2016] (K2-ABC) as measures of discrepancy in ABC, along with SA-ABC [Fearnhead and Prangle, 2012]. The models with which we conduct experiments were chosen to cover a range application domains, namely: ecology, finance, and public

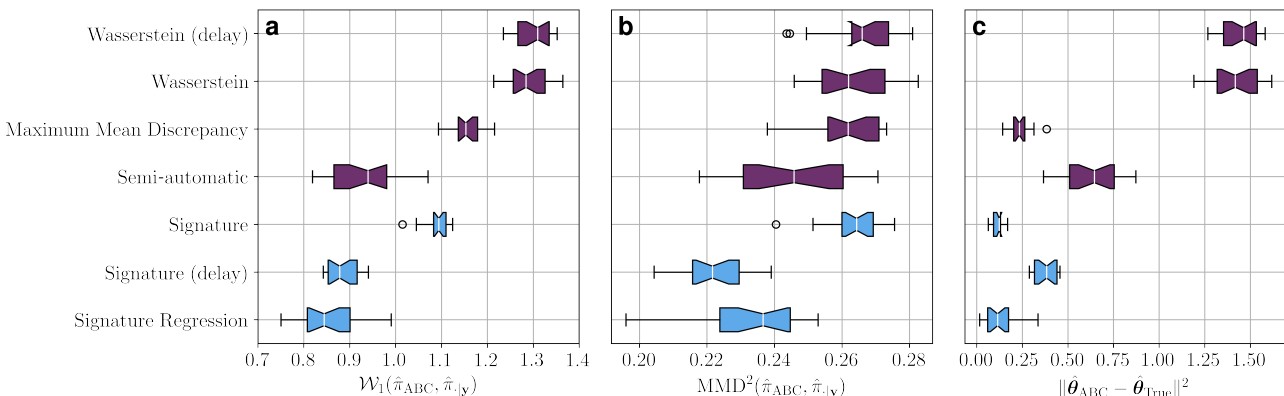

Figure 2: (Ricker model) (**a**) Wasserstein distances and (**b**) maximum mean discrepancies between the posteriors recovered from each ABC method and an approximate ground truth from particle Markov chain Monte Carlo (pMCMC). (**c**) Squared distances between the means of the ABC posteriors and the posterior mean from pMCMC. Our methods are shown in blue.

health/epidemiology. These models were also chosen due to the fact that approximate ground-truth posteriors are readily available via standard MCMC techniques, permitting a proper evaluation of the methods' performance in the posterior inference task. Finally, they were also chosen for the variety of outputs they produce: chaotic, integer-valued time-series in the first example; non-stationary, real-valued sequences in the second; and continuous-time, variable-length sequences of multivariate and irregularly spaced points in the final case. Further details on the experiments we present below, along with additional results, are provided in Appendix D.

## 4.1 IMPLEMENTATION DETAILS

For all distances, we sample from the ABC posterior using a simple REJ-ABC scheme as outlined in Algorithm 1 and, unless stated otherwise, use $N = 10^5$ and $M = 10^3$. While other, more sophisticated schemes exist, we choose this to facilitate a simple and transparent comparison of the different distance measures. To assess the quality of the recovered posteriors, we compute the 1-Wasserstein distance and an unbiased estimate of the maximum mean discrepancy (MMD) between the approximate ground truth posteriors $\hat{\pi}_{\cdot|\mathbf{y}}$ and empirical posteriors $\hat{\pi}_{ABC}$. In both cases, smaller values indicate a closer match to the approximate ground truth. To estimate the MMD between posteriors, we use a Gaussian RBF kernel with scale parameter chosen with the median heuristic [Briol et al., 2019]. For S-ABC and W-ABC, we also report results obtained by applying a (lag-1) *delay* transformation to the time series before distance computations, which acts on a time series $\mathbf{x}$ as

$$(\mathbf{x}_{t_1}, \mathbf{x}_{t_2}, \ldots, \mathbf{x}_{t_n}) \mapsto$$
$$((\mathbf{x}_{t_1}, \mathbf{x}_{t_2}), (\mathbf{x}_{t_2}, \mathbf{x}_{t_3}), \ldots, (\mathbf{x}_{t_{n-1}}, \mathbf{x}_{t_n})).$$

Such a transformation was considered in Bernton et al. [2019] for time series data, and may improve the accur-

acy of the ABC posteriors in practical, non-asymptotic settings. Results obtained with such a transformation are indicated by a "(delay)" suffix. All other implementation details are provided in Appendix D.2. Code for the experiments is available at `https://github.com/joelnmdyer/SignatureABC`.

## 4.2 THE RICKER MODEL

The Ricker model is a simple model of ecological dynamics that exhibits chaotic behaviour and has an intractable likelihood function. The state of the model, which tracks the size $N_t \in \mathbb{R}_{\geq 0}$ of a population over discrete time steps $t = 1, \ldots, n$, evolves as

$$\log N_{t+1} = \log r + \log N_t - N_t + \sigma \epsilon_t, \qquad (17)$$

where $r > 0$ is a growth parameter and $\epsilon_t \sim \mathcal{N}(0, 1)$. Following Wood [2010], we assume Poissonian observations

$$\mathbf{y}_t \sim \text{Po}(\phi N_t) \in \mathbb{N}, \qquad (18)$$

where $\phi > 0$ is a scale parameter. We assume the task of recovering the posterior distribution for $\boldsymbol{\theta} = (\log r, \phi, \sigma)$ given a time series of length $n = 50$, $\mathbf{y} = (\mathbf{y}_1, \mathbf{y}_2, \ldots, \mathbf{y}_n) \sim p_{\boldsymbol{\theta}^*}$ with $\boldsymbol{\theta}^* = (4, 10, 0.3)$. We take $N_0 = 1$. We further assume the following independent, uniform priors for each parameter:

$$\log r \sim \mathcal{U}(3, 8), \quad \phi \sim \mathcal{U}(0, 20), \quad \sigma \sim \mathcal{U}(0, 0.6). \quad (19)$$

For SA-ABC, the hand-crafted summary statistics we use are those proposed in Wood [2010], and consist of: the autocovariances to lag 5; the mean; the number of zeros in the sequence; the coefficients of the regression $\mathbf{x}_{t+1}^{0.3} = \beta_1 \mathbf{x}_t^{0.3} + \beta_2 \mathbf{x}_t^{0.6} + \epsilon_t$ for error term $\epsilon_t$; and the coefficients of the cubic regression of the ordered differences $\mathbf{x}_t - \mathbf{x}_{t-1}$ on their observed values.

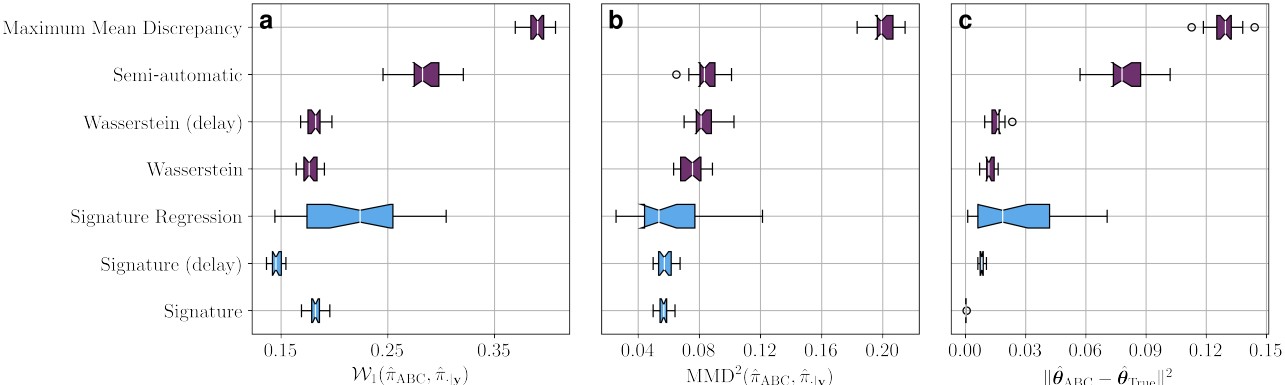

Figure 3: (Geometric Brownian motion) (**a**) Wasserstein distances and (**b**) maximum mean discrepancies between the posteriors obtained with each ABC method and an approximate ground truth from Metropolis-Hastings (MH). (**c**) Squared distances between the means of the ABC posteriors and the posterior mean from MH. Our methods are coloured blue.

In Figure 2, we show boxplots for the Wasserstein distances and MMDs between samples from the ABC posteriors – denoted with $\hat{\pi}_{\mathrm{ABC}}$ – and samples from an approximation of the true posterior obtained using pMCMC (Andrieu et al. [2010]; see Appendix D.3 for details), which we denote with $\hat{\pi}_{.|\mathbf{y}}$. We also show boxplots for the Euclidean distances between the ABC posterior means and the pMCMC posterior mean. These boxplots are all obtained by running the ABC procedure 20 times with the same observed dataset but different seeds for the ABC procedure.

From this, we see that the signature-based methods tend to produce better performance across all three metrics considered. In more detail, the estimate of the approximate ground truth posterior obtained with the signature-based methods are more accurate than K2-ABC and W-ABC, as reflected in the Wasserstein distances and MMDs. For S-ABC, this performance gap is enhanced with the additional application of the lag-1 delay transformation, which is indicated with suffix "(delay)" in Figure 2. No such improvement is observed when this transformation is applied to competing methods. We note that SA-ABC performs particularly well in this example, as a consequence of its use of hand-crafted summary statistics developed specifically for this simulation model. However, the potential power of our signature-based methods is demonstrated by the fact that SR-ABC is able to outperform SA-ABC in all three metrics, despite the latter using summary statistics carefully engineered by experts. Finally, we observe more accurate estimates of the true posterior mean using our signature-based methods than using W-ABC and SA-ABC. The posterior mean estimates from S-ABC without the delay transformation and SR-ABC are also more accurate than those of MMD, further evidencing the usefulness of our signature-based methods.

## 4.3 GEOMETRIC BROWNIAN MOTION

Geometric Brownian motion (GBM) is a stochastic differential equation widely used in mathematical finance to model a stock price $x_t$ evolving with time $t$ according to

$$\mathrm{d}x_t = \mu x_t \mathrm{d}t + \sigma x_t \mathrm{d}W_t, \qquad (20)$$

where $\mu$ is the percentage drift, $\sigma$ is the volatility, and $W_t$ is a Brownian motion. With $\epsilon_i \sim \mathcal{N}(0, 1)$, this model permits an exact discretisation for $i = 1, 2, \ldots, n-1$ as

$$\log\left(\mathbf{x}_{i\Delta t}/\mathbf{x}_{(i-1)\Delta t}\right) = \left(\mu - \frac{1}{2}\sigma^2\right)\Delta t + \sigma\sqrt{\Delta t}\,\epsilon_i, \quad (21)$$

which implicitly defines the model $p_{\boldsymbol{\theta}}$ from which we simulate. We fix $\mathbf{x}_0 = 10$, $n = 100$, and $\Delta t = 1/(n-1)$, and simulate the dynamics over the interval $[0, 1]$.

We consider the task of recovering the posterior for $\boldsymbol{\theta} = (\mu, \sigma)$ given an observation $\mathbf{y} = (\mathbf{y}_0, \mathbf{y}_{\Delta t}, \mathbf{y}_{2\Delta t}, \ldots, \mathbf{y}_{(n-1)\Delta t}) \sim p_{\boldsymbol{\theta}^*}$ with $\boldsymbol{\theta}^* = (0.2, 0.5)$. We assume independent, uniform priors $\mu \sim \mathcal{U}(-1, 1)$, $\sigma \sim \mathcal{U}(0.2, 2)$. Inference is amenable to standard, exact likelihood-based Bayesian techniques such as Metropolis-Hastings (MH) sampling using the transition density implied by (21), enabling a comparison against an approximate ground truth posterior. For SA-ABC, we follow Fearnhead and Prangle [2012] and regress the parameters $\boldsymbol{\theta}$ onto the first, second, third, and fourth powers of summary statistics of the time series. Specifically, we take the first, second, third, and fourth powers of the variance and lag-1 and -2 autocorrelations of the increments of the log time series, $\log\left(\mathbf{x}_{i\Delta t}/\mathbf{x}_{(i-1)\Delta t}\right)$, since these are informative of the parameters being inferred.

In Figure 3, we show boxplots for the Wasserstein distances and MMDs between the different ABC posteriors and the approximate ground truth posterior obtained with MH, in addition to the Euclidean distance between the ABC posterior

Table 1: Median (1st quartile–3rd quartile) Performance Metrics for Section 4.4. Best Performance in **Bold**.

| Method | $\mathcal{W}_1(\hat{\pi}_{\text{ABC}}, \hat{\pi}_{\cdot|\mathbf{y}})$ $(\times 10^{-3})$ | $\text{MMD}^2(\hat{\pi}_{\text{ABC}}, \hat{\pi}_{\cdot|\mathbf{y}})$ $(\times 10^{-2})$ | $\|\hat{\boldsymbol{\theta}}_{\text{ABC}} - \hat{\boldsymbol{\theta}}_{\text{True}}\|^2$ $(\times 10^{-5})$ |
|---|---|---|---|
| Signature ABC | **4.8 (4.3–5.6)** | **6.1 (4.1–7.8)** | **0.32 (0.22–0.43)** |
| Wasserstein ABC | 7.3 (6.4–7.7) | 7.6 (6.8–9.1) | 0.46 (0.29–0.72) |
| K2-ABC | 520.4 (519.8–521.3) | 34.44 (34.39–34.49) | 27036 (26947–27126) |

means and the MH posterior mean. The boxplots were generated by repeating the rejection ABC procedure for each distance measure with 20 different random seeds. We see that the signature-based methods once again produce lower Wasserstein distances and MMDs between their ABC posteriors and the MH posterior. Indeed, S-ABC with the lag-1 delay transformation uniformly dominates the non-signature methods across all three metrics.

## 4.4 IRREGULAR, MULTIVARIATE SEQUENCES: GENERALISED STOCHASTIC EPIDEMICS

The signature method naturally allows for inference with multivariate and/or irregularly spaced time series. To demonstrate this, we consider a generalised stochastic epidemic model [Kypraios, 2007], which simulates the spread of an infection through a fixed population of $Z$ individuals. Individuals are initially susceptible, may become infected, and subsequently recover without the possibility of reinfection. The dynamics of the model are determined by parameters $\beta$ and $\gamma$, which control the rate of infection and recovery according to the following transition probabilities:

$$P_I := \mathbb{P}\left((\delta X_t, \delta Y_t) = (-1, 1) \mid \mathcal{H}_t\right) = \beta X_t Y_t \delta t + o(\delta t),$$
$$P_R := \mathbb{P}\left((\delta X_t, \delta Y_t) = (0, -1) \mid \mathcal{H}_t\right) = \gamma Y_t \delta t + o(\delta t),$$
$$\mathbb{P}\left((\delta X_t, \delta Y_t) = (0, 0) \mid \mathcal{H}_t\right) = 1 - (P_I + P_R) + o(\delta t),$$

where $X_t$ and $Y_t$ are the number of susceptible and infected individuals at time $t \in [0, T]$, respectively, and $\mathcal{H}_t$ is a sigma-algebra generated by the process up until time $t$. These three transition probabilities thus capture infection, recovery, and an absence of activity, respectively.

We consider the problem of recovering the posterior density for $\boldsymbol{\theta} = (\beta, \gamma)$ given observations of the infections and recoveries occurring in the observation period $[0, 50]$ in a system of $Z = 100$ individuals. For every simulation, the epidemic begins with one infected individual at time $t = 0$. We generate "empirical" data at parameters $\boldsymbol{\theta}^* = (10^{-2}, 10^{-1})$, and assume priors $\beta \sim \Gamma(\lambda_\beta, \nu_\beta)$ and $\gamma \sim \Gamma(\lambda_\gamma, \nu_\gamma)$, with concentration and rate parameters $\lambda_\beta = 0.1$, $\nu_\beta = 2$, $\lambda_\gamma = 0.2$, and $\nu_\gamma = 0.5$. It can be shown [Kypraios, 2007] that this prior is conjugate for the model, leading to a tractable posterior density; further details are provided in Appendix D.6. Thus, samples can be drawn from the exact posterior for a given dataset simulated by this model.

We simulate the model using the Gillespie algorithm [Gillespie, 1977], such that the lengths of the simulated sequences, and the spacing between points in the sequences, are random. Operationally, the model is simulated as followed: given that an infection/recovery event occurred at time $t$, the time $\Delta t$ until the next event is simulated as $\Delta t \sim \text{Exp}(1/R_t)$ where $R_t = \beta X_t Y_t + \gamma Y_t$, and the event is chosen to be an infection (resp. recovery) event with probability $\beta X_t Y_t / R_t$ (resp. $\gamma Y_t / R_t$).

We show in Table 1 the median and first and third quartiles for the Wasserstein distances and MMDs between samples from W-ABC, S-ABC, and K2-ABC posteriors to samples from the exact posterior. To obtain these approximate posteriors, we run Algorithm 1 with the same observed time series with $N = 10^5$ and $M = 100$ for 20 different seeds for the ABC procedure. We also show the same for the squared distance between the posterior means and the exact posterior mean. Contour plots obtained by running the inference procedure at these 20 different seeds for the ABC procedure and pooling the best $M$ distances from each (giving 2000 samples) are shown in Figure 4, along with samples from the exact posterior. The MMD performed especially poorly in this experiment; we thus omit samples from the K2-ABC posterior in Figure 4 for clarity.

From this, we see that the natural notion of distance between multivariate and irregularly sampled time series data of different lengths, enabled by the use of path signatures, manifests as better recovery of both the true posterior distribution and the true posterior mean in this example, in which the Wasserstein distances and MMDs between posteriors and Euclidean distances between posterior means for S-ABC are generally lower than those obtained using the Wasserstein distance and the MMD is distances in ABC.

## 4.5 COMPUTATIONAL COMPLEXITY AND COST

Evaluating the signature kernel for two streams $\mathbf{y} \in \mathcal{X}^n$ and $\mathbf{x} \in \mathcal{X}^m$ with $\mathcal{X} = \mathbb{R}^d$ has complexity that is linear in $d$ and linear in the product $nm$ [Salvi et al., 2021]. This is likewise the case for MMD, which has complexity $\mathcal{O}\left(n^2\right)$ [Park et al., 2016], and compares favourably with the Wasserstein distance, which in multivariate settings is known to scale poorly with $n$. Bernton et al. [2019], for example, note costs of order $n^3$ when the Hungarian algorithm is used to solve the assignment problem. Alternative algorithms with favourable performance are an active area of research.

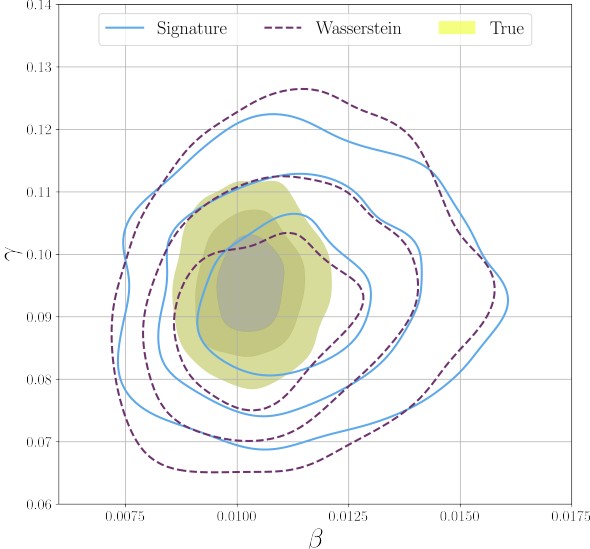

Figure 4: (Stochastic epidemic) Contour plot of the joint posterior density recovered with the Wasserstein distance (dashed purple lines) and Signature ABC (solid blue lines), and samples from the true posterior (filled yellow contours).

While the complexity of signature evaluations compares favourably to alternative distance measures, we observe in our experiments that our signature-based approaches tended to incur larger computational costs with current implementations; see Table 2. These increased costs may not be an inherent feature of signature-based methods, however: research on signature methods in machine learning and computational statistics is active and relatively nascent, and it is plausible that more efficient implementations of signature computations will emerge with time. Additionally, techniques for reducing the computational burden of the signature-based methods we introduce can be employed, such as the Nyström method [Williams and Seeger, 2000] or random Fourier features [Rahimi and Recht, 2007, Yang et al., 2012] in SR-ABC, and the truncated signature kernel [Király and Oberhauser, 2019] in both S-ABC and SR-ABC. While we have not experimented with these cost-reduction techniques in the current work, future practical implementations of S-ABC and SR-ABC may incorporate such approximations, each of which have been implemented in, e.g., the KSig package. Such approximations introduce further hyperparameters such as the truncation degree, however, which must be tuned; in the case of SR-ABC, this can be done with cross-validation, but it is less clear how this might be done for S-ABC. Furthermore, in the case of S-ABC, too severe a truncation may destroy the asymptotic results presented in Proposition 1. Nevertheless, these are avenues that can be explored in future work in order to reduce the computational burden of these methods.

## 5   CONCLUSION

In this paper, we introduced two novel approaches – Signature ABC and Signature Regression ABC – to performing approximate Bayesian computation with time series simulation models. Each method relies on the path signature – an object that is fundamental to the theory of controlled differential equations and rough paths – which is associated with the path traversed by a sequence of data points. In particular, we make use of the signature kernel to construct and compute discrepancies between time series data arising in ABC settings without manually contriving summary statistics.

We show that the natural notion of distance between time series to which such an approach leads generates an ABC posterior that converges to the ground-truth posterior as the ABC tolerance parameter reduces to 0. To illustrate our proposed methods, we present multiple examples of Bayesian inference tasks in which our approaches outperform existing techniques that are common in the approximate Bayesian inference literature; indeed, in each experiment we consider, at least one signature-based method uniformly dominates competing methods across all three of the metrics considered in this paper. We furthermore demonstrate that our methods are applicable to more complex settings than univariate time series, for example simulators generating complex multivariate and irregularly sampled sequences.

While we have compared the different distance measures using a basic rejection algorithm in this paper in the interest of a simple and transparent comparison, we note that our proposed methods can be embedded within other more sophisticated sampling algorithms, for example MCMC or sequential Monte Carlo methods. Additionally for the Signature Regression ABC method, there is the possibility of incorporating mechanisms for generating more accurate regression results, for example using a pilot run to determine regions of non-negligible posterior mass as described in Fearnhead and Prangle [2012]. This may allow for improved approximations to the true posterior density. With respect to the choice of S-ABC vs. SR-ABC: whether one should be preferred over the other depends to a large extent on what is of interest to the experimenter, given that semi-automatic approaches to ABC were originally motivated by the desire to accurately recover point estimates of interest [Fearnhead and Prangle, 2012] while other ABC methods aim to accurately approximate the full posterior. Additionally, we observe empirically that SR-ABC and SA-ABC seem to exhibit a somewhat larger variation in performance over non-semi-automatic approaches (see, e.g., Figures 2, 3, and 6), which may be a manifestation of the additional stochasticity introduced in training a regression model prior to posterior construction.

Table 2: Approximate Average CPU Times (in seconds) for each ABC Approach. (Simulation Budgets and Hardware Availability are Constant).

| Experiment | Method | | | | | | |
|---|---|---|---|---|---|---|---|
| | S-ABC | S-ABC (delay) | SR-ABC | W-ABC | W-ABC (delay) | SA-ABC | K2-ABC |
| Ricker | $2 \times 10^2$ | $2 \times 10^2$ | $2 \times 10^4$ | $6 \times 10^1$ | $8 \times 10^1$ | $10^2$ | $4 \times 10^1$ |
| GBM | $10^3$ | $10^4$ | $6 \times 10^4$ | $4 \times 10^3$ | $4 \times 10^3$ | $9 \times 10^2$ | $4 \times 10^3$ |
| GSE | $10^4$ | – | – | $2 \times 10^2$ | – | – | $10^5$ |

## Acknowledgements

The authors are grateful to Horatio Boedihardjo, Lajos Gergely Gyurko, Zacharia Issa, Terry Lyons, James Morrill, Harald Oberhauser, and Cristopher Salvi for their comments, feedback, and helpful discussions. JD was supported by the EPSRC Centre For Doctoral Training in Industrially Focused Mathematical Modelling (EP/L015803/1) in collaboration with Improbable. JD was also supported by the EPSRC grant EP/W002949/1 and The Alan Turing Institute under the EPSRC grant EP/N510129/1.

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

# Approximate Bayesian Computation with Path Signatures
## (Supplementary Material)

**Joel Dyer**[1,2]         **Patrick Cannon**[3]         **Sebastian M. Schmon**[3]

[1]Department of Computer Science, University of Oxford
[2]Institute for New Economic Thinking, University of Oxford
[3]No affiliation

## A   A SIMPLE EXAMPLE OF THE WASSERSTEIN CURVE MATCHING DISTANCE FOR TIME SERIES DATA

In Section 2, we discuss that the Wasserstein curve matching distance will generally permit permutations of the elements within the two time series that are being compared, and that this limits the suitability of the method for time series data – a data type for which the ordering of the data is in general of ultimate importance.

As a simple example of this feature of the behaviour of the Wasserstein curve matching distance, consider the following two time series with elements observed at times $t = 0, 1, 2$:

$$\mathbf{x} = (\mathbf{x}_0, \mathbf{x}_1, \mathbf{x}_2) = (1, 3, 2)$$
$$\mathbf{y} = (\mathbf{y}_0, \mathbf{y}_1, \mathbf{y}_2) = (5, 1, 4).$$

The matrix of pairwise distances for each element of $\mathbf{x}$ with each element of $\mathbf{y}$ under the distance in Equation (6) is then

$$M(\mathbf{x}, \mathbf{y}) = \begin{bmatrix} 4 & 0 & 3 \\ 2 & 2 & 1 \\ 3 & 1 & 2 \end{bmatrix} + \lambda \begin{bmatrix} 0 & 1 & 2 \\ 1 & 0 & 1 \\ 2 & 1 & 0 \end{bmatrix}. \tag{22}$$

With the choice of $\lambda = 1$, the minimal-cost assignment of elements in $\mathbf{x}$ to elements in $\mathbf{y}$ will be to match $\mathbf{x}_0$ with $\mathbf{y}_1$, $\mathbf{x}_1$ with $\mathbf{y}_0$, and $\mathbf{x}_2$ with $\mathbf{y}_2$, with a total associated cost of 2. This example demonstrates that this approach continues to treat the elements of the time series as fundamentally exchangeable, despite the incorporation of information regarding their ordering in the distance measure used. Such an approach has limited suitability to time series data for this reason.

## B   PATH SIGNATURES

### B.1   FURTHER BACKGROUND ON PATH SIGNATURES

In the main text, we introduced path signatures as maps from $h \in BV_{[0,T]}(\mathcal{H})$ to elements of $\prod_{m \geq 0} \mathcal{H}^{\otimes m}$ with finite norm, where the $m$th term of the signature takes value

$$S_m(h) := \int_0^T \mathrm{d}h^{\otimes m} := \int_0^T \int_0^t \mathrm{d}h^{\otimes(m-1)} \otimes \mathrm{d}h_t = \int \cdots \int_{0 \leq t_1 < \cdots < t_m \leq T} \mathrm{d}h_{t_1} \otimes \cdots \otimes \mathrm{d}h_{t_m}. \tag{23}$$

This introduction to path signatures, provided in Section 2, is quite general. To introduce signatures more completely, it is instructive to consider the special case of a smooth, finite-dimensional path $x : [0, T] \to \mathcal{H}$ with $\mathcal{H} = \mathbb{R}^d$. The depth-$m$

term of the path signature for $x$ can then be written as

$$S_m(x) = \int_0^T dx^{\otimes m} = \underset{0 \leq t_1 < \cdots < t_m \leq T}{\int \cdots \int} \left( \left.\frac{dx}{dt}\right|_{t_1} \otimes \cdots \otimes \left.\frac{dx}{dt}\right|_{t_m} \right) dt_1 \ldots dt_m, \tag{24}$$

where for $a \in \mathbb{R}^{\alpha_1 \times \cdots \times \alpha_k}$ and $b \in \mathbb{R}^{\beta_1 \times \cdots \times \beta_l}$ the tensor product $\otimes$ operates as $(a_{i_1,\ldots,i_k}, b_{j_1,\ldots,j_l})$ $\mapsto a_{i_1,\ldots,i_k} b_{j_1,\ldots,j_l}$ and the integrals can be taken in the Riemann-Stieltjes sense.

**Remark 1.** *Since we have assumed our paths to be of bounded variation, the integrals above can be understood as the Riemann-Stieljes integrals with respect to $h$. When the underlying path is not smooth, the integrals are taken to be stochastic or rough path integrals [Chevyrev and Oberhauser, 2018]. For example, in the case of Brownian motion in $\mathbb{R}^d$, the integrals are stochastic and can be taken in the Stratonovich sense. For a larger class of stochastic processes, rough path theory [Lyons et al., 2007] provides an integration theory that enables the computation of the terms in the signature. As we will discuss later, this work considers throughout only linear interpolations between points in time series, so all paths considered here are of finite variation.*

Path signatures are thus infinite sequences of statistics for path-valued random variables capturing information regarding the order of observations along, and the interaction between, different channels of the path. They are grounded in the theory of (CDEs) and stochastic analysis, and appear in the solutions of CDEs and (SDEs) as obtained through a procedure analogous to Picard iterations for ordinary differential equations.

To see this, we follow Lyons et al. [2007] and let $V$ and $W$ be two Banach spaces, $B : V \rightarrow \mathbf{L}(W, W)$ be a bounded linear map – where $\mathbf{L}(W, W)$ denotes the space of bounded linear mappings from $W \rightarrow W$ — and $h : [0, T] \rightarrow V$ be a continuous path of bounded variation. Consider the following set of linear equations:

$$dg_t = Bg_t \, dh_t, \quad g_0 \in W \tag{25}$$
$$d\phi_t = B\phi_t \, dh_t, \quad \phi_0 \in \mathbf{L}(W, W). \tag{26}$$

Here, $Bg_t \, dh_t$ is taken to mean $\{B(dh_t)\}(g_t)$ while $B\phi_t \, dh_t$ is $B(dh_t) \circ \phi_t$. By applying the aforementioned iterative procedure to recover the solution $\phi_t$ to (26), we obtain

$$\phi_t = \sum_{m \geq 0} B^{\otimes m} \int_0^t dh^{\otimes m}, \tag{27}$$

in which we see that the signature terms, Equation (23), appear in the summand. The solution to (25) is then obtained from the flow $\phi_t$ as $g_t = \phi_t(h_0)$. Similarly, a solution to the following linear SDE driven by Brownian motion $W$,

$$dY_t = A(Y_t) \circ dW_t, \quad Y_0 = y_0$$

for some linear operator $A$, can be obtained as

$$Y_t = \sum_{m \geq 0} A^{\otimes m} S_{m,[0,t]}(W) \, y_0,$$

where $S_{m,[0,t]}(W)$ is the order-$m$ tensor in the signature of $W_t$ over interval $[0, t]$ and the integrals are taken in the Stratonovich sense [Lyons et al., 2007, Section 3.3.2]. As we have seen here, signatures arise naturally as good approximations to solutions of CDEs and SDEs, and accurately describe the response of systems such as that of Equations (25)-(26) to an input signal $h$, where the inclusion of terms of increasing order further refine the approximate solution. The above sums, such as in Equation (27), converges as a result of the factorial rate of decay of the terms in the signature:

**Proposition 2** (Proposition 2.2, Lyons et al. [2007]). *Let $V$ be a Banach space and $h \in BV_{[0,T]}(V)$. Then, for each $m \geq 0$,*

$$\left\| \int_0^T dh^{\otimes m} \right\|_{V^{\otimes m}} \leq \frac{\|h\|_{1-\text{var}}}{m!}. \tag{28}$$

**Remark 2.** *The signature of a univariate path consists only of powers of the difference between the final and initial points in the stream [see e.g. Chevyrev and Kormilitzin, 2016, Example 5]. Therefore in practice one always considers paths in at least two dimensions. This can always be achieved by including the observation time as a channel in the path.*

## B.2 INVARIANCES

Further properties of the signature include its translation and reparameterisation invariance (Section 2.2.2, Lyons et al. [2007]; Theorem 3.4.2, Geng [2015]):

**Proposition 3.** *Let $h \in BV_{[0,T]}(\mathcal{H})$, $a \in \mathcal{H}$, and $\psi : [0, T] \to [0, T]$ a non-decreasing surjection. Then $\mathrm{Sig}(h+a) = \mathrm{Sig}(h)$ and $\mathrm{Sig}(h \circ \psi) = \mathrm{Sig}(h)$.*

In this way, signatures are able to factor out nuisance and potentially infinite-dimensional symmetries where this is beneficial. However, when such invariances are disadvantageous, they can easily be destroyed with two extremely simple preprocessing techniques: *time-augmentation*, in which the path $(t, h_t)$ is instead considered, and *basepoint augmentation*, in which $h_0 = c$ for some fixed constant $c \in \mathcal{H}$ is enforced for all paths under consideration.

Another more interesting invariance property results from the signature's inability to identify regions of the path in which, informally speaking, a retracing of the path occurs [Chen, 1958, Hambly and Lyons, 2010, Boedihardjo et al., 2016]; that is, for example, paths of the form $a \star b \star \overleftarrow{b} \star c$ for $a, b, c \in BV_{[0,T]}(\mathcal{H})$, where $\star$ denotes concatenation and $\overleftarrow{b}$ is the path $b$ "run-backwards". Paths in which such retracings occur are referred to as *tree-like equivalent* to their reduced paths such that, for example, $a \star b \star \overleftarrow{b} \star c \sim_t a \star c$, where $\sim_t$ denotes tree-like equivalence. While this phenomenom has previously been studied in more specific cases [Chen, 1958, Hambly and Lyons, 2010], the most general form of this invariance property is provided by Boedihardjo et al. [2016], a special case of which may be stated as follows:

**Theorem 1** (Hambly and Lyons [2010], Boedihardjo et al. [2016]). *Let $V$ be a Banach space and $h, g \in BV_{[0,T]}(V)$. Then $\mathrm{Sig}(h) = \mathrm{Sig}(g)$ iff $h \sim_t g$.*

In the real world, however, tree-like equivalent paths are rare and can straightforwardly be avoided by considering only time-augmented paths $h : [0, T] \to \mathcal{H} \times [0, T]$, $t \mapsto (t, h_t)$. Such a transformation ensures that the path is injective, meaning no partial retracing can occur at any point along the path. This, along with their universal nonlinearity property, demonstrates that signatures are powerful and faithful representations of paths and are, essentially, an injective feature map for path-valued random variables. Signatures are therefore an appealing option for performing inference for stochastic process simulators.

## B.3 SHUFFLE-PRODUCT PROPERTY

The terms of the path signature exhibit a so-called *shuffle-product* property:

**Theorem 2** (Theorem 2.29, Lyons et al. [2007]). *Let $h \in BV_{[0,T]}(\mathcal{H})$. Then*

$$\int_0^T \mathrm{d}h^{\otimes m} \otimes \int_0^T \mathrm{d}h^{\otimes m'} = \sum_\sigma \sigma \left( \int_0^T \mathrm{d}h^{\otimes (m+m')} \right),$$

*where the sum is taken over all order shuffles, defined as*

$$\{\sigma : \sigma \text{ is a permutation of } \{1, \dots, m + m'\} \text{ with } \sigma(1) < \cdots < \sigma(m), \sigma(m+1) < \cdots < \sigma(m+m')\}.$$

*$\sigma$ then acts on $\mathcal{H}^{\otimes(m+m')}$ as $\sigma(e_{i_1} \otimes \cdots \otimes e_{i_{m+m'}}) = e_{\sigma(i_1)} \otimes \cdots \otimes e_{\sigma(i_{m+m'})}$.*

## B.4 FUNCTION APPROXIMATION CAPABILITIES

We state informally in the main text that signatures enjoy a universal nonlinearity property. This may be stated more formally as follows:

**Theorem 3** (Appendix A.2, Kiŕaly and Oberhauser [2019]). *Let $\mathcal{K}$ be a compact set of non-tree-like (see Appendix B.2) paths of bounded variation, and $C(\mathcal{K}, \mathbb{R})$ be the space of continuous, real-valued function on $\mathcal{K}$. Then the space of linear functionals on signatures of paths in $\mathcal{K}$ is dense in $C(\mathcal{K}, \mathbb{R})$; that is, for any $f \in C(\mathcal{K}, \mathbb{R})$ and any $\varepsilon > 0$, there exists an $L \in \bigoplus_{m \geq 0} \mathcal{H}^{\otimes m}$ such that*

$$\sup_{h \in \mathcal{K}} \left| f(h) - L\{\mathrm{Sig}(h)\} \right| < \varepsilon.$$

This is a consequence of the *shuffle product* property of signatures (see Appendix B.3 above) and the Stone-Weierstrass theorem. (An issue that arises in the application of the classical Stone-Weierstrass theorem in this context is that the space of interest to us – $BV_{[0,T]}(\mathcal{H})$ – is not locally compact. The classical Stone-Weierstrass theorem therefore cannot strictly be applied here. However, Chevyrev and Oberhauser [2018] demonstrate that a Stone-Weierstrass result exists by equipping the space of continuous bounded real-valued functions on $BV_{[0,T]}(\mathcal{H})$ with an appropriate topology. See Chevyrev and Oberhauser [2018] for details.)

## B.5 ADDITIONAL PRE-PROCESSING

Prior to lifting the sequence to a path in $\mathcal{H}$, and depending on the nature of the data at hand, it is sometimes appropriate to apply a transformation to the data: certain transformations may enable the signature to represent information in the stream more conveniently for the learning task at hand. A large set of such transformations have been proposed in the literature on inference using path signatures; see Morrill et al. [2020] for a recent summary and comparison of many of these. Here, we describe two such pre-signature transformations that we will use in this paper.

**Cumulative sum**  Recall from Figure 1 that the depth 1 signature terms correspond to the increment along the path, and that a subset of the depth 2 terms correspond to the areas above and below the curve. For certain data types, for example non-negative binary or spiking data, the data may not be well-characterised by these terms by default. In such cases it can be beneficial to consider instead the cumulative sum of the observations [Kiŕaly and Oberhauser, 2019], which can intuitively be thought of as propagating information from earlier in the sequence to later in the stream, more readily exhibiting the structure of the stream. The effect of this can be to shift information into lower order terms in the signature, for example the increments (depth 1 terms).

**Delay transformation**  A transformation that is common in time series analysis is a delay transformation, for example the lag-1 delay transformation:

$$(\mathbf{x}_{t_1}, \mathbf{x}_{t_2}, \ldots, \mathbf{x}_{t_n}) \mapsto ((\mathbf{x}_{t_1}, \mathbf{x}_{t_2}), (\mathbf{x}_{t_2}, \mathbf{x}_{t_3}), \ldots, (\mathbf{x}_{t_{n-1}}, \mathbf{x}_{t_n})). \tag{29}$$

Applying this transformation before applying the signature may help to encode temporal features of the time series.

### B.5.1  Augmentations

As noted previously, two augmentations can be applied to remove the signature's translation and reparameterisation invariance properties:

**Time augmentation**  In this transformation, the uniformly increasing time index $0 = t_1 < t_2 < \cdots < t_n = T$ is added as a channel in the sequence:

$$(\mathbf{x}_{t_1}, \mathbf{x}_{t_2}, \ldots, \mathbf{x}_{t_n}) \mapsto ((t_1, \mathbf{x}_{t_1}), (t_2, \mathbf{x}_{t_2}), \ldots, (t_n, \mathbf{x}_{t_n})), \tag{30}$$

denoting the times at which the points in the series occurred.

**Basepoint augmentation**  With this transformation, all sequences are enforced to assume a common but otherwise arbitrary initial value. This can be achieved by simply concatenating an arbitrary constant value to the beginning of each sequence.

**Lead-lag transformation**  This transformation operates on a sequence $\mathbf{x} = (\mathbf{x}_{t_1}, \mathbf{x}_{t_2}, \ldots, \mathbf{x}_{t_n})$ as follows:

$$(\mathbf{x}_{t_1}, \mathbf{x}_{t_2}, \ldots, \mathbf{x}_{t_n}) \mapsto ((\mathbf{x}_{t_1}, \mathbf{x}_{t_1}), (\mathbf{x}_{t_1}, \mathbf{x}_{t_2}), (\mathbf{x}_{t_2}, \mathbf{x}_{t_2}), \ldots, (\mathbf{x}_{t_{n-1}}, \mathbf{x}_{t_n}), (\mathbf{x}_{t_n}, \mathbf{x}_{t_n})). \tag{31}$$

Under this transformation, the number of channels in the sequence doubles, and the sequence length increases from $n$ to $2n - 1$. Applying this transformation enables the signature to emphasise certain properties of the path such as the quadratic variation and the Lévy area when combined with the cumulative sum [Gyurk, 2014, Chevyrev and Kormilitzin, 2016]. For datasets for which these quantities are believed to be important, applying the lead-lag transformation may be appropriate.

## C  PROOF OF PROPOSITION 1

In this section, we denote the space of piecewise linear paths of bounded variation in $\mathcal{H}$ over time interval $[0, T]$ with $\mathcal{P}_{[0,T]}(\mathcal{H})$. We will furthermore abuse notation slightly by letting $\kappa(\mathbf{x}, \cdot) \in \mathcal{P}_{[0,T]}(\mathcal{H})$ denote the path in Equation (13), i.e.

the linear interpolation of the lifted points $(\kappa(\mathbf{x}_{t_1}, \cdot), \kappa(\mathbf{x}_{t_2}, \cdot), \ldots, \kappa(\mathbf{x}_{t_n}, \cdot))$, while denoting the feature map for $\mathbf{x}_t$ with $\kappa(\mathbf{x}_t, \cdot) \in \mathcal{H}$. Finally, we will take $k(\mathbf{x}, \cdot) := \mathrm{Sig}(\mathbf{x})$ to mean the signature of the piecewise linear, $\mathcal{H}$-valued path $\kappa(\mathbf{x}, \cdot)$, while $\mathrm{Sig}(g)$ denotes the signature of a path $g \in BV_{[0,T]}(\mathcal{H})$.

We demonstrate that the discrepancy measure in Equation (14) satisfies the conditions specified in Proposition 3.1 of Bernton et al. [2019], which gives a statement on the convergence of ABC posteriors to the true posterior under certain regularity conditions on the simulator's likelihood function as $\varepsilon \to 0$. A specific case of the statement is as follows:

**Proposition 4** (Proposition 3.1, Bernton et al. [2019]). *Let $\mathcal{X} := \mathbb{R}^d$, $\mathbf{y} = (\mathbf{y}_1, \ldots, \mathbf{y}_n) \in \mathcal{X}^n$, and $\mathcal{D} : \mathcal{X}^n \times \mathcal{X}^n \to \mathbb{R}_{\geq 0}$ be a non-negative distance measure on $\mathcal{X}^n$. Suppose $p_{\boldsymbol{\theta}}(\mathbf{x})$ is the continuous density (with respect to the Lebesgue measure) associated with simulated data $\mathbf{x} \in \mathcal{X}^n$ and that*

$$\sup_{\boldsymbol{\theta} \in \Theta \setminus \mathcal{N}_{\Theta}} p_{\boldsymbol{\theta}}(\mathbf{x}) < \infty,$$

*where $\mathcal{N}_{\Theta}$ is a set such that $\pi(\boldsymbol{\theta}) = 0 \, \forall \boldsymbol{\theta} \in \mathcal{N}_{\Theta}$. Suppose further that there exists $\bar{\varepsilon} > 0$ such that*

$$\sup_{\boldsymbol{\theta} \in \Theta \setminus \mathcal{N}_{\Theta}} \sup_{\mathbf{z} \in \mathcal{A}^{\bar{\varepsilon}}} p_{\boldsymbol{\theta}}(\mathbf{z}) < \infty,$$

*where $\mathcal{A}^{\bar{\varepsilon}} := \{\mathbf{z} : \mathcal{D}(\mathbf{y}, \mathbf{z}) \leq \bar{\varepsilon}\}$. Suppose that $\mathcal{D}$ is continuous. If $\mathcal{D}(\mathbf{y}, \mathbf{z}) = 0$ iff $\mathbf{y} = \mathbf{z}$, keeping $\mathbf{y}$ fixed, then for any measurable $\mathbf{B} \subset \Theta$,*

$$\lim_{\varepsilon \to 0} \int_{\mathbf{B}} \pi_{\varepsilon}(\boldsymbol{\theta} \mid \mathbf{y}) \, \mathrm{d}\boldsymbol{\theta} = \int_{\mathbf{B}} \pi(\boldsymbol{\theta} \mid \mathbf{y}) \, \mathrm{d}\boldsymbol{\theta}. \tag{32}$$

Therefore, provided that the stated regularity conditions on the simulator's likelihood function are met, showing that the distance function in Equation (14) is continuous and injective is sufficient to show that the S-ABC posterior converges to the true posterior as $\varepsilon \to 0$. These requirements are indeed met under the assumptions of Proposition 4 and under additional benign conditions:

**Proposition 5.** *Let $\mathcal{X} := \mathbb{R}^d$, $\mathbf{y} = (\mathbf{y}_1, \ldots, \mathbf{y}_n) \in \mathcal{X}^n$ be the fixed real-world dataset, and $\mathbf{x}$ be a simulated dataset. Assume both $\mathbf{y}$ and $\mathbf{x}$ are time- and basepoint-augmented, and that $\kappa : \mathcal{X} \times \mathcal{X} \to \mathbb{R}$ is a uniformly bounded kernel with continuous, injective canonical feature map. Let $\mathcal{D}(\mathbf{y}, \cdot)$ be as in Equation (14), i.e.,*

$$\mathcal{D}(\mathbf{y}, \cdot) := \rho \{\mathrm{Sig}(\mathbf{y}), \cdot\} \circ \mathrm{Sig} \circ \kappa : \mathcal{X}^n \to \mathbb{R}_{\geq 0}, \quad \mathbf{x} \mapsto \|\mathrm{Sig}(\mathbf{y}) - \mathrm{Sig}(\mathbf{x})\|^2 \tag{33}$$

*consisting of lifting the sequence $\mathbf{x} \in \mathcal{X}^n$ to a piecewise linear path in $\mathcal{H}$, before computing the squared distance between its signature and $\mathrm{Sig}(\mathbf{y})$. Then this map is uniformly continuous.*

We will proceed by noting that each constituent map in the above operation is a continuous map, and the result follows since compositions of continuous maps are continuous.

**Lemma 1.** *Let $\mathcal{X}^n$ be the space of length-$n$ basepoint-augmented sequences in $\mathcal{X} = \mathbb{R}^d$ and $\mathbf{x}, \mathbf{z} \in \mathcal{X}^n$. Then the one-variation*

$$\|\mathbf{x}\|_{1-\mathrm{var}} = \sum_{i=1}^{n-1} \|\mathbf{x}_{i+1} - \mathbf{x}_i\|_{\mathcal{X}} \tag{34}$$

*is a norm on $\mathcal{X}^n$.*

*Proof.* The triangle inequality follows immediately as a result of the triangle inequality for the norm on $\mathcal{X}$:

$$\begin{aligned}
\|\mathbf{x} + \mathbf{z}\|_{1-\mathrm{var}} &= \sum_{i=1}^{n-1} \|(\mathbf{x}_{i+1} + \mathbf{z}_{i+1}) - (\mathbf{x}_i + \mathbf{z}_i)\|_{\mathcal{X}} \\
&\leq \sum_{i=1}^{n-1} \|\mathbf{x}_{i+1} - \mathbf{x}_i\|_{\mathcal{X}} + \|\mathbf{z}_{i+1} - \mathbf{z}_i\|_{\mathcal{X}} \\
&= \|\mathbf{x}\|_{1-\mathrm{var}} + \|\mathbf{z}\|_{1-\mathrm{var}}.
\end{aligned}$$

Absolute homogeneity is also immediate:

$$\|s\mathbf{x}\|_{1-\mathrm{var}} = \sum_{i=1}^{n-1} \|s\mathbf{x}_{i+1} - s\mathbf{x}_i\|_{\mathcal{X}} = |s| \sum_{i=1}^{n-1} \|\mathbf{x}_{i+1} - \mathbf{x}_i\|_{\mathcal{X}} = |s| \|\mathbf{x}\|_{1-\mathrm{var}}.$$

Finally, since the streams are basepoint-augmented, meaning $\mathbf{x}_1 = 0$ for all $\mathbf{x} \in \mathcal{X}^n$, we have that $\|\mathbf{x}\|_{1-\mathrm{var}} = 0$ iff $\mathbf{x} = (0, 0, \ldots, 0)$:

$$\|\mathbf{x}\|_{1-\mathrm{var}} = 0 \implies \|\mathbf{x}_{i+1} - \mathbf{x}_i\|_{\mathcal{X}} = 0 \; \forall \, i = 1, \ldots, n-1 \implies \mathbf{x}_i = \mathbf{x}_1 = 0 \; \forall \, i.$$

$\square$

We next show that lifting length-$n$ basepoint-augmented sequences in $\mathcal{X}$ to sequences in $\mathcal{H}$ is continuous if the canonical feature map $\phi$ associated with $\kappa$ is itself continuous:

**Lemma 2.** *Let $\mathcal{X}^n$ be the space of length-$n$ basepoint-augmented sequences in $\mathcal{X} = \mathbb{R}^d$, $\mathbf{x}, \mathbf{z} \in \mathcal{X}^n$, and $\phi : \mathcal{X} \to \mathcal{H}$ be the canonical feature map associated with kernel $\kappa$ with RKHS $\mathcal{H}$. Assume $\phi$ is continuous. Then the map $\mathbf{x} \mapsto \kappa(\mathbf{x}, \cdot)$ – where $\kappa(\mathbf{x}, \cdot)$ is the linear interpolation of the points $(\phi(\mathbf{x}_1), \ldots, \phi(\mathbf{x}_n))$ in $\mathcal{H}$ – is continuous in the one-variation topology.*

*Proof.* By Lemma 1, the one-variation is a norm on length-$n$ basepoint-augmented sequences in $\mathcal{X}$. We will proceed by showing that the one-variation is an equivalent norm to the 1-product norm, defined as

$$\|\mathbf{x}\|_{\mathcal{X}^n} := \sum_{i=1}^{n} \|\mathbf{x}_i\|_{\mathcal{X}}, \tag{35}$$

which induces the product topology on $\mathcal{X}^n$. By showing this, we will have the following implications: from the definition of the 1-product norm,

$$\|\mathbf{x} - \mathbf{z}\|_{\mathcal{X}^n} < \tilde{\delta} \implies \|\mathbf{x}_i - \mathbf{z}_i\|_{\mathcal{X}} < \tilde{\delta} \text{ also;} \tag{36}$$

by continuity of $\phi$, we have that $\forall \, \tilde{\epsilon} > 0, \exists \, \tilde{\delta} > 0$ such that

$$\|\mathbf{x}_i - \mathbf{z}_i\|_{\mathcal{X}} < \tilde{\delta} \implies \|\phi(\mathbf{x}_i) - \phi(\mathbf{z}_i)\|_{\mathcal{H}} < \tilde{\epsilon}; \tag{37}$$

and that choosing $\tilde{\epsilon} = \epsilon/2(n-1)$ for any $\epsilon > 0$ means that ensuring $\|\phi(\mathbf{x}_i) - \phi(\mathbf{z}_i)\|_{\mathcal{H}} < \tilde{\epsilon}$ for all $i$ means

$$
\begin{aligned}
\|\kappa(\mathbf{x}, \cdot) - \kappa(\mathbf{z}, \cdot)\|_{1-\mathrm{var}} &= \sum_{i=1}^{n-1} \|\{\phi(\mathbf{x}_{i+1}) - \phi(\mathbf{z}_{i+1})\} - \{\phi(\mathbf{x}_i) - \phi(\mathbf{z}_i)\}\|_{\mathcal{H}} \\
&\leq \sum_{i=1}^{n-1} \|\phi(\mathbf{x}_{i+1}) - \phi(\mathbf{z}_{i+1})\|_{\mathcal{H}} + \|\phi(\mathbf{x}_i) - \phi(\mathbf{z}_i)\|_{\mathcal{H}} \\
&< 2(n-1)\tilde{\epsilon} \\
&= \epsilon.
\end{aligned}
\tag{38}
$$

We therefore have the following chain of implications: for every $\epsilon > 0$ there is a $\tilde{\delta} > 0$ such that

$$
\begin{aligned}
\|\mathbf{x} - \mathbf{z}\|_{\mathcal{X}^n} < \tilde{\delta} \implies \|\mathbf{x}_i - \mathbf{z}_i\|_{\mathcal{X}} < \tilde{\delta} &\implies \|\phi(\mathbf{x}_i) - \phi(\mathbf{z}_i)\|_{\mathcal{H}} < \tilde{\epsilon} \\
&\implies \|\kappa(\mathbf{x}, \cdot) - \kappa(\mathbf{z}, \cdot)\|_{1-\mathrm{var}} < \epsilon.
\end{aligned}
\tag{39}
$$

It therefore suffices to show that for any $\tilde{\delta} > 0$ there is a $\delta > 0$ such that $\|\mathbf{x} - \mathbf{z}\|_{1-\mathrm{var}} < \delta \implies \|\mathbf{x} - \mathbf{z}\|_{\mathcal{X}^n} < \tilde{\delta}$, which by this chain of implications would imply that $\forall \, \epsilon > 0, \exists \, \delta > 0$ such that $\|\mathbf{x} - \mathbf{z}\|_{1-\mathrm{var}} < \delta \implies \|\kappa(\mathbf{x}, \cdot) - \kappa(\mathbf{z}, \cdot)\|_{1-\mathrm{var}} < \epsilon$. This follows immediately, since $\| \cdot \|_{1-\mathrm{var}}$ and $\| \cdot \|_{\mathcal{X}^n}$ are norms on finite-dimensional vector spaces, and are thus equivalent norms. In particular, we have that $\|\mathbf{x}\|_{\mathcal{X}^n} \leq \|\mathbf{x}\|_{1-\mathrm{var}}/c$, such that for all $\tilde{\delta} > 0$, we have that

$$\|\mathbf{x} - \mathbf{z}\|_{1-\mathrm{var}} < \delta := c\tilde{\delta} \implies \|\mathbf{x} - \mathbf{z}\|_{\mathcal{X}^n} < \tilde{\delta}, \tag{40}$$

and so we are done. $\square$

We consider next the continuity of the signature map for piecewise linear paths of bounded variation in $\mathcal{H}$. For such paths, the signature truncated at degree 1 is a multiplicative functional with bounded variation (see Lyons et al. [2002, Section 3.1.2]) and, consequently, a special case of Lyons et al. [2002, Theorem 3.1.3] applies:

**Lemma 3.** *Let $V$ be a Banach space, $x, z \in BV_{[0,T]}(V)$ be two bounded variation paths in $V$, and $\tau$ be a constant such that*

$$\tau \geq 2 \left\{ 1 + \sum_{r=3}^{\infty} \left( \frac{2}{r-2} \right)^2 \right\}.$$

*If $\varphi$ is a constant such that*

$$\|x\|_{1-\mathrm{var}}, \|z\|_{1-\mathrm{var}} \leq \frac{\varphi}{\tau} \qquad \text{and} \qquad \|x - z\|_{1-\mathrm{var}} \leq \chi \frac{\varphi}{\tau}$$

*for some $\chi > 0$, then for all $m \geq 1$*

$$\|S_m(x) - S_m(z)\|_{V^{\otimes m}} \leq \frac{\chi}{\tau} \cdot \frac{\varphi^m}{m!}. \tag{41}$$

An immediate consequence of this is that the signature map is continuous in the 1-variation topology for bounded variation paths in Banach spaces:

**Corollary 1.** *Let $\mathcal{H}$ be a Hilbert space, $x, z \in BV_{[0,T]}(\mathcal{H})$ be two bounded variation paths in $\mathcal{H}$, and $\tau$ be as in Lemma 3. If $\varphi$ is a constant such that*

$$\|x\|_{1-\mathrm{var}}, \|z\|_{1-\mathrm{var}} \leq \frac{\varphi}{\tau} \qquad \text{and} \qquad \|x - z\|_{1-\mathrm{var}} \leq \chi \frac{\varphi}{\tau}$$

*for some $\chi > 0$, then*

$$\|\mathrm{Sig}(x) - \mathrm{Sig}(z)\| \leq \frac{\chi}{\tau} \exp\left( \frac{\varphi^2}{2} \right).$$

*Proof.* By definition of the norm on $\prod_{m \geq 0} \mathcal{H}^{\otimes m}$,

$$\|\mathrm{Sig}(x) - \mathrm{Sig}(z)\| = \sqrt{\sum_{m \geq 0} \|S_m(x) - S_m(z)\|_{\mathcal{H}^{\otimes m}}^2}$$

$$= \sqrt{0 + \sum_{m \geq 1} \|S_m(x) - S_m(z)\|_{\mathcal{H}^{\otimes m}}^2} \qquad (S_0(x) = 1 \,\forall x \in BV_{[0,T]}(\mathcal{H}))$$

$$\leq \sqrt{\sum_{m \geq 1} \frac{\chi^2}{\tau^2} \cdot \left( \frac{\varphi^m}{m!} \right)^2} \qquad \text{(from (41) above)}$$

$$= \frac{\chi}{\tau} \sqrt{\sum_{m \geq 1} \frac{(\varphi^2)^m}{(m!)^2}}$$

$$\leq \frac{\chi}{\tau} \sqrt{\sum_{m \geq 1} \frac{(\varphi^2)^m}{m!}} \qquad \text{(smaller denominator)}$$

$$\leq \frac{\chi}{\tau} \exp\left( \frac{\varphi^2}{2} \right). \qquad \text{(convergent series)}$$

$\square$

We show next that the map $\rho\left(\mathrm{Sig}(\mathbf{y}), \cdot\right) : \prod_{m \geq 0} \mathcal{H}^{\otimes m} \to \mathbb{R}_{\geq 0}$, $s \mapsto \|\mathrm{Sig}(\mathbf{y}) - s\|^2$ is continuous. To do so, we make use of the following result:

**Lemma 4.** *Let $\kappa$ be a uniformly bounded kernel i.e. one for which $\sup_{x \in \mathcal{X}} \sqrt{\kappa(x, x)} < \infty$, and let $\kappa(\mathbf{x}, \cdot) \in \mathcal{P}_{[0,T]}(\mathcal{H})$ be a $\mathcal{H}$-valued piecewise linear path with knots at $\kappa(\mathbf{x}_i, \cdot), i = 1, \ldots, n$, and $\mathrm{Sig}(\mathbf{x})$ its signature. Then*

$$\sup_{\mathbf{x} \in \mathcal{X}^n} \|\mathrm{Sig}(\mathbf{x})\| < \infty. \tag{42}$$

*Proof.* For all $\mathbf{x} \in \mathcal{X}^n$, we have

$$\|\kappa(\mathbf{x}, \cdot)\|_{1-\mathrm{var}} = \sum_{i=1}^{n-1} \|\kappa(\mathbf{x}_{i+1}, \cdot) - \kappa(\mathbf{x}_i, \cdot)\|_{\mathcal{H}} \qquad \text{(piecewise linear)}$$

$$\leq \sum_{i=1}^{n-1} \|\kappa(\mathbf{x}_{i+1}, \cdot)\|_{\mathcal{H}} + \|\kappa(\mathbf{x}_i, \cdot)\|_{\mathcal{H}} \qquad \text{(triangle inequality)}$$

$$= \sum_{i=1}^{n-1} \sqrt{\kappa(\mathbf{x}_{i+1}, \mathbf{x}_{i+1})} + \sqrt{\kappa(\mathbf{x}_i, \mathbf{x}_i)} \qquad \text{(reproducing property)}$$

$$\leq 2(n-1) \sup_{z \in \mathcal{X}} \sqrt{\kappa(z, z)}. \qquad \text{($\kappa$ bounded)}$$

Let $v := 2(n-1) \sup_{z \in \mathcal{X}} \sqrt{\kappa(z, z)}$. Then $\forall \mathbf{x} \in \mathcal{X}^n$,

$$\|\mathrm{Sig}(\mathbf{x})\| \leq \left\{ \sum_{m=0}^{\infty} \frac{(\|\kappa(\mathbf{x}, \cdot)\|_{1-\mathrm{var}}^2)^m}{(m!)^2} \right\}^{\frac{1}{2}} \qquad \text{(Proposition 2)}$$

$$\leq \left\{ \sum_{m=0}^{\infty} \frac{(v^2)^m}{m!} \right\}^{\frac{1}{2}}$$

$$= e^{\frac{v^2}{2}}, \qquad \text{(exponential series)}$$

where in the first inequality we make use of the factorial decay property of signatures. We obtain the result by taking the supremum over $\mathcal{X}^n$:

$$\sup_{\mathbf{x} \in \mathcal{X}^n} \|\mathrm{Sig}(\mathbf{x})\| \leq e^{\frac{v^2}{2}} < \infty.$$

$\square$

**Lemma 5.** *Let $\kappa$ be a uniformly bounded kernel i.e. one for which $\sup_{z \in \mathcal{X}} \sqrt{\kappa(z, z)} < \infty$, and let $\kappa(\mathbf{y}, \cdot) \in \mathcal{P}_{[0,T]}(\mathcal{H})$ be the observed $\mathcal{H}$-valued piecewise linear path with $\mathrm{Sig}(\mathbf{y})$ its signature. Denote the signature kernel as*

$$k(\mathbf{x}, \mathbf{z}) = \langle \mathrm{Sig}(\mathbf{x}), \mathrm{Sig}(\mathbf{z}) \rangle \tag{43}$$

*Then the distance function*

$$\rho\left(\mathrm{Sig}(\mathbf{y}), \cdot\right) : \prod_{m \geq 0} \mathcal{H}^{\otimes m} \to \mathbb{R}_{\geq 0}, \quad s \mapsto \|s - \mathrm{Sig}(\mathbf{y})\|^2 \tag{44}$$

*is Lipschitz continuous in $s$.*

*Proof.*

$$\left| \mathcal{D}(\mathbf{y}, \mathbf{x}) - \mathcal{D}(\mathbf{y}, \mathbf{z}) \right| = \left| \|\mathrm{Sig}(\mathbf{x}) - \mathrm{Sig}(\mathbf{y})\|^2 - \|\mathrm{Sig}(\mathbf{z}) - \mathrm{Sig}(\mathbf{y})\|^2 \right|$$

$$= \left| k(\mathbf{x}, \mathbf{x}) - k(\mathbf{z}, \mathbf{z}) + 2 \left( k(\mathbf{z}, \mathbf{y}) - k(\mathbf{x}, \mathbf{y}) \right) \right|$$

$$\leq \left| k(\mathbf{x}, \mathbf{x}) - k(\mathbf{z}, \mathbf{z}) \right| + 2 \left| k(\mathbf{z}, \mathbf{y}) - k(\mathbf{x}, \mathbf{y}) \right| \qquad \text{(triangle inequality)}$$

Considering the first of these terms and making use of the reproducing property and symmetry of $k$:

$$\left| k(\mathbf{x}, \mathbf{x}) - k(\mathbf{z}, \mathbf{z}) \right| = \left| k(\mathbf{x}, \mathbf{x}) - k(\mathbf{x}, \mathbf{z}) + k(\mathbf{z}, \mathbf{x}) - k(\mathbf{z}, \mathbf{z}) \right|$$

$$= \left| \langle k(\mathbf{x}, \cdot), k(\mathbf{x}, \cdot) - k(\mathbf{z}, \cdot) \rangle + \langle k(\mathbf{z}, \cdot), k(\mathbf{x}, \cdot) - k(\mathbf{z}, \cdot) \rangle \right|$$

$$\leq \left| \langle k(\mathbf{x}, \cdot), k(\mathbf{x}, \cdot) - k(\mathbf{z}, \cdot) \rangle \right| + \left| \langle k(\mathbf{z}, \cdot), k(\mathbf{x}, \cdot) - k(\mathbf{z}, \cdot) \rangle \right|$$

$$\leq \left( \|\mathrm{Sig}(\mathbf{x})\| + \|\mathrm{Sig}(\mathbf{z})\| \right) \cdot \|\mathrm{Sig}(\mathbf{x}) - \mathrm{Sig}(\mathbf{z})\|,$$

where in the penultimate and final lines we use the triangle inequality and the Cauchy-Schwarz inequality twice, respectively. Considering now the second term:

$$\left| k(\mathbf{z}, \mathbf{y}) - k(\mathbf{x}, \mathbf{y}) \right| = \left| \langle \mathrm{Sig}(\mathbf{y}), \mathrm{Sig}(\mathbf{x}) - \mathrm{Sig}(\mathbf{z}) \rangle \right|$$
$$\leq \| \mathrm{Sig}(\mathbf{y}) \| \, \| \mathrm{Sig}(\mathbf{x}) - \mathrm{Sig}(\mathbf{z}) \| , \qquad \text{(Cauchy-Schwartz)}$$

where in the first line we use the definition and symmetry of the inner product. Putting the two terms together and using Lemma 5, we have

$$\left| \mathcal{D}(\mathbf{y}, \mathbf{x}) - \mathcal{D}(\mathbf{y}, \mathbf{z}) \right| \leq \left( \| \mathrm{Sig}(\mathbf{x}) \| + \| \mathrm{Sig}(\mathbf{z}) \| + 2 \| \mathrm{Sig}(\mathbf{y}) \| \right) \| \mathrm{Sig}(\mathbf{x}) - \mathrm{Sig}(\mathbf{z}) \|$$
$$\leq 4 e^{\frac{v^2}{2}} \| \mathrm{Sig}(\mathbf{x}) - \mathrm{Sig}(\mathbf{z}) \|$$

where $v$ is as in Lemma 4. Thus $\rho\left(\mathrm{Sig}(\mathbf{y}), \cdot\right)$ is Lipschitz continuous. $\qquad\square$

We finally arrive at the conclusion:

*Proof of Proposition 5.* Compositions of continuous maps are continuous, and each of the constituent maps are continuous from the Lemmas and Corollaries presented above. $\qquad\square$

Injectivity of the signature map is also guaranteed under these conditions:

**Proposition 6.** *Let $\mathcal{X} := \mathbb{R}^d$, $\mathbf{x}, \mathbf{y} \in \mathcal{X}^n$. Assume both $\mathbf{x}$ and $\mathbf{y}$ are time- and basepoint-augmented, and that $\kappa : \mathcal{X} \times \mathcal{X} \to \mathbb{R}$ is a uniformly bounded kernel with continuous, injective canonical feature map. Then $\mathrm{Sig}(\mathbf{x}) = \mathrm{Sig}(\mathbf{y})$ iff $\mathbf{x} = \mathbf{y}$.*

*Proof.* Obtaining a signature from a length-$n$ data stream $\mathbf{x}$ entails: (1) lifting the points $\mathbf{x}_i$ in $\mathbf{x}$ to the RKHS $\mathcal{H}$ associated with $\kappa$ as $\kappa(\mathbf{x}_i, \cdot)$; (2) applying a linear interpolation to obtain a piecewise linear $\mathcal{H}$-valued path $\kappa(\mathbf{x}, \cdot)$; and (3) finally taking the signature of $\kappa(\mathbf{x}, \cdot)$. To show injectivity of this composite map, it suffices to show injectivity of each of these three steps since the composition of injective maps is injective.

(1) is trivially injective, due to the assumed injectivity of $\kappa$. (2) is by definition injective for a length-$n$ sequence in $\mathcal{H}$. To show injectivity of (3), we note that time-augmentation of the sequences, along with injectivity of $\kappa$, ensure that the lifted paths are injective, such that no tree-like equivalence is observed between the interpolated paths in $\mathcal{H}$. Time-augmentation further makes the signature sensitive to parameterisation, removing its parameterisation invariance property. Uniform boundedness of $\kappa$ ensures that $\kappa(\mathbf{x}, \cdot)$ is of bounded variation, such that $\kappa(\mathbf{x}, \cdot) \in \mathcal{P}_{[0,T]}(\mathcal{H})$. To see this, note that for a piecewise linear path $\kappa(\mathbf{x}, \cdot)$,

$$\| \kappa(\mathbf{x}, \cdot) \|_{1-\mathrm{var}} = \sum_{i=1}^{n-1} \| \kappa(\mathbf{x}_{i+1}, \cdot) - \kappa(\mathbf{x}_i, \cdot) \|_{\mathcal{H}} \leq 2(n-1) \sup_{\mathbf{z} \in \mathcal{X}} \sqrt{\kappa(\mathbf{z}, \mathbf{z})} < \infty,$$

where we have used the reproducing property of $\kappa$ and the triangle inequality. Finally, since basepoint augmentation makes the signature sensitive to paths that differ only by translations, the desired result follows from Theorem 1. $\qquad\square$

# D  FURTHER EXPERIMENTAL DETAILS

## D.1  SIGNATURE REGRESSION ABC

For SR-ABC, we proceed as follows:

(a) fit a kernel ridge regression model using training data $\{\mathbf{x}^{(i)}, \boldsymbol{\theta}^{(i)}\}_{i=1}^R \sim p_{\boldsymbol{\theta}}(\mathbf{x}) \, \pi(\boldsymbol{\theta})$. This amounts to solving the following optimisation problem for each of the $p$ components $j = 1, \ldots, p$ of the $\{\boldsymbol{\theta}^{(i)}\}_{i=1}^R$:

$$\min_{\hat{\boldsymbol{\theta}}_j \in \mathcal{H}_k} \sum_{i=1}^R \left\{ \boldsymbol{\theta}_j^{(i)} - \hat{\boldsymbol{\theta}}_j\left(\mathbf{x}^{(i)}\right) \right\}^2 + \alpha \| \hat{\boldsymbol{\theta}}_j \|_{\mathcal{H}_k}^2, \qquad (45)$$

where $k$ is the signature kernel, $\mathcal{H}_k$ is the RKHS associated with $k$, $\hat{\boldsymbol{\theta}}_j$ is – by the Representer Theorem – a function of the form

$$\hat{\boldsymbol{\theta}}_j(\mathbf{x}) = \sum_{i=1}^{R} \boldsymbol{\omega}_i^{(j)} k(\mathbf{x}, \mathbf{x}^{(i)}) \tag{46}$$

with

$$\boldsymbol{\omega}^{(j)} = (G + \alpha I_R)^{-1} \boldsymbol{\psi}^{(j)}, \qquad G_{mn} = k(\mathbf{x}^{(m)}, \mathbf{x}^{(n)}), \qquad \boldsymbol{\psi}^{(j)} = \begin{bmatrix} \boldsymbol{\theta}_j^{(1)} \\ \boldsymbol{\theta}_j^{(2)} \\ \vdots \\ \boldsymbol{\theta}_j^{(R)} \end{bmatrix}, \qquad I_R = \mathrm{diag}(1, 1, \dots, 1) \in \mathbb{R}^{R \times R},$$

and $\alpha \geq 0$ is a regularisation parameter;

(b) summarise the observation $\mathbf{y}$ and all future simulations $\mathbf{x} \sim p_{\boldsymbol{\theta}}$ using this trained kernel ridge regression model, i.e. use

$$\mathbf{s}(\mathbf{x}) = \begin{bmatrix} \hat{\boldsymbol{\theta}}_1(\mathbf{x}) \\ \hat{\boldsymbol{\theta}}_2(\mathbf{x}) \\ \vdots \\ \hat{\boldsymbol{\theta}}_p(\mathbf{x}) \end{bmatrix}; \tag{47}$$

(c) use the squared difference between the summaries of $\mathbf{y}$ and $\mathbf{x}$ as the measure of discrepancy between simulation and observation,

$$\rho\{\mathbf{s}(\mathbf{y}), \mathbf{s}(\mathbf{x})\} = \|\mathbf{s}(\mathbf{y}) - \mathbf{s}(\mathbf{x})\|_2^2. \tag{48}$$

## D.2 FURTHER IMPLEMENTATION DETAILS

For all signature kernel computations, we use the `sigkernel` package [Salvi et al., 2021] and we normalise the time series by dividing by the range of the simulation output when this is known or, when this is unknown, with the expected range of the training set of size $R = 300$ for SR-ABC or $R = 300$ samples from the prior predictive distribution for S-ABC.

Unless stated otherwise, we remove the translation invariance and reparameterisation-invariance properties of the signature – discussed in Appendix B.2 – by applying basepoint and time-augmentations to all time series in every experiment.

Unless stated otherwise, we take $\kappa$ to be a Gaussian RBF kernel with scale hyperparameter $\sigma$. To tune $\sigma$ and the regularisation hyperparameter for SR-ABC, we perform a grid search with 5-fold cross-validation on the training set. For S-ABC, we use the median of all pairwise Euclidean distances between points in the observation $\mathbf{y}$ for $\sigma$, although we note that other approaches could be taken, such as using the same method as for SR-ABC.

Both SA-ABC and SR-ABC require training data; for both we use $R = 300$ training examples $\{\mathbf{x}^{(j)}, \boldsymbol{\theta}^{(j)}\}_{j=1}^{R} \sim p_{\boldsymbol{\theta}}(\mathbf{x})\pi(\boldsymbol{\theta})$. When $\pi(\cdot)$ has bounded support, we normalise the parameters $\{\boldsymbol{\theta}^{(i)}\}_{i=1}^{R}$ in the training set with the range of the prior in each dimension. We also tune the bandwidth parameter for the Gaussian RBF kernel employed in the MMD distance for K2-ABC using the median of the pairwise absolute differences between observations in $\mathbf{y}$, as recommended by Park et al. [2016].

In all experiments, "Wasserstein" and "W-ABC" indicates the use of the 1-Wasserstein distance with curve matching, which as described in Section 2 is a method for using the Wasserstein distance for time series recommended in Bernton et al. [2019], in the rejection ABC sampling scheme. To determine the $\lambda$ coefficient, we follow the guidance of Thorpe et al. [2017] and choose

$$\lambda \simeq \frac{V}{T}, \tag{49}$$

where $V$ is the expected vertical range and $T$ is the length of the time interval over which observations are made, in order to balance the effects of vertical and horizontal transport. Where the value of $V$ is not apparent *a priori*, we estimate it using $R = 300$ samples from the prior predictive distribution. Distances are computed using the Python Optimal Transport package [Flamary et al., 2021].

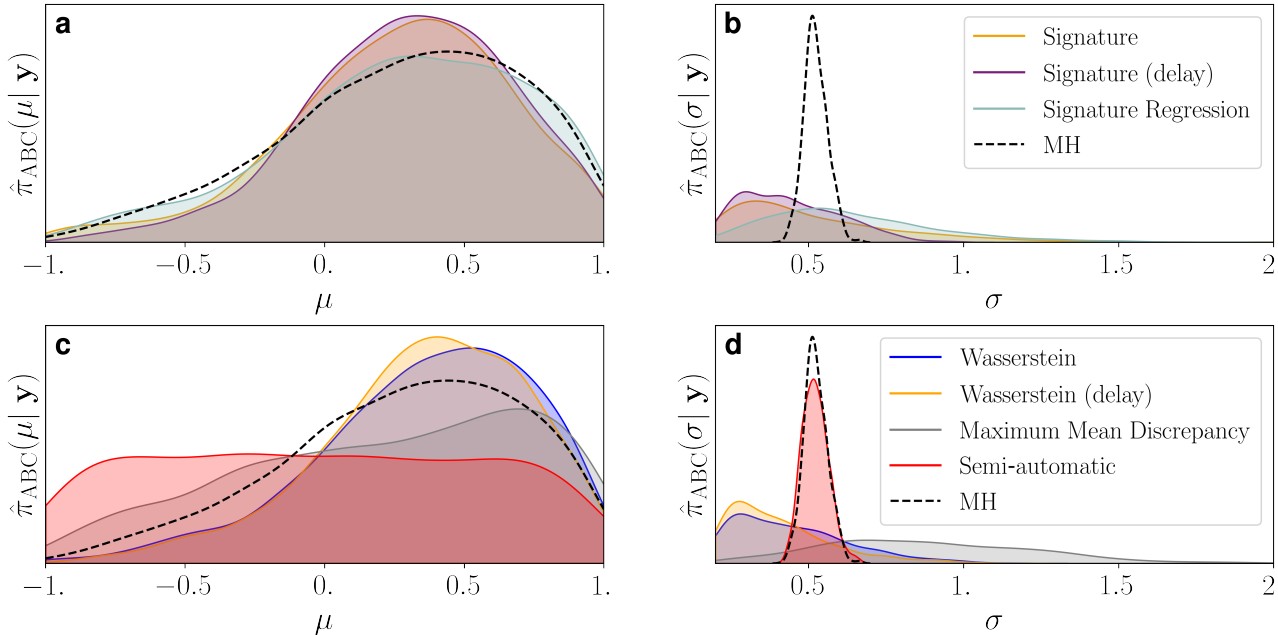

Figure 5: (Geometric Brownian motion) Examples of marginal posterior distributions recovered using each distance function and the approximate ground-truth posterior recovered with a Metropolis-Hastings (MH) random walk. Panels **a** and **b** show the marginal posteriors recovered using our signature methods (S-ABC and Signature regression ABC (SR-ABC)) and the approximate ground-truth posterior (MH). Panels **c** and **d** show the marginal posteriors recovered using the Wasserstein curve matching distance (Wasserstein), double kernel ABC (K2-ABC) (maximum mean discrepancy (MMD)), and semi-automatic approximate Bayesian computation (ABC) with powers of the variance and lag-1 and -2 autocorrelations of the increments of the log time series as regressors (semi-automatic ABC (SA-ABC)).

## D.3 REFERENCE POSTERIORS USING MCMC

**Metropolis-Hastings** For the geometric Brownian motion (GBM) and Brock & Hommes models, we obtain samples from the ground truth posterior using Metropolis-Hastings (MH). We follow the guidelines of Schmon and Gagnon [2022] and use a multivariate normal proposal, for which we estimate the covariance matrix using a pilot run. We subsequently tune the MH algorithm according to Schmon and Gagnon [2022, Table 1] and run the MH for $10^5$ steps, keeping a thinned subset of $10^3$ samples as our baseline.

**Particle MCMC** To obtain samples from the ground truth posterior of the Ricker model we employ particle Markov chain Monte Carlo (pMCMC) using a simple bootstrap particle filter. We follow the guidelines of Schmon et al. [2021], first estimating the posterior covariance in a shorter prior run and then tuning the random walk proposal as well as the particle filter. PMCMC commonly exhibits worse convergence behaviour than standard MH and hence we run the algorithm for $2 \times 10^5$ iterations eventually retaining a thinned subset of $10^3$ samples as our baseline.

## D.4 FURTHER EXPERIMENTAL DETAILS: THE RICKER MODEL

The time series generated by the Ricker model tend to consist of many zero terms, with occasional spikes. For this reason, we use the cumulative sum pre-signature transformation (see Appendix B.5) for Signature ABC (S-ABC), which is a common transformation for spiking data such as medical data [Morrill et al., 2019]. In our experiments, we also found that W-ABC and K2-ABC benefited from this transform and were not competitive without it. We therefore also report the results obtained with W-ABC and K2-ABC with this cumulative sum transform applied.

## D.5 FURTHER EXPERIMENTAL RESULTS: GEOMETRIC BROWNIAN MOTION

We show in Figure 5 the marginal posteriors recovered using the Metropolis-Hastings (MH) approximation (see Appendix D.3 for details) and the true likelihood function, along with the approximate posteriors obtained using the rejection sampling scheme in Algorithm 1 and each of the distance measures considered. The suffix "(delay)" once again indicates that the lag-1 delay transformation was applied. From this, we see that and SR-ABC and S-ABC track the shape of the approximate ground truth marginal posterior generated by MH for $\mu$ more closely than all other methods, and that the marginal distribution for $\sigma$ concentrates in the neighbourhood of the approximate ground-truth marginal posterior for $\sigma$. This is in contrast to, for example, the MMD, which is overly dispersed and biased for $\sigma$.

In this example, SA-ABC has been able to very accurately approximate the marginal density for $\sigma$ as a consequence of the informative set of summary statistics provided to this method. However, SA-ABC has experienced difficulty recovering the shape of the marginal density for $\mu$, despite the provided summary statistics also being informative of this parameter. The fact that the signature- and Wasserstein-based methods are able to outperform SA-ABC, despite the advantage the latter has been afforded, illustrates the potential power of these methods in cases where the model structure is too complex to easily derive summary statistics that are informative of the parameters.

## D.6 FURTHER EXPERIMENTAL DETAILS: GENERALISED STOCHASTIC EPIDEMICS

For the priors reported in the main text, the posterior density can be written as

$$\pi(\beta, \gamma \mid \mathbf{I}, \mathbf{R}) \propto \beta^{\lambda_\beta + n_I - 2} \exp\left\{ -\beta \left( \int_{\phi_1}^T X_t Y_t \, \mathrm{d}t + \nu_\beta \right) \right\} \gamma^{\lambda_\gamma + n_R - 1} \exp\left\{ -\gamma \left( \int_{\phi_1}^T Y_t \, \mathrm{d}t + \nu_\gamma \right) \right\}, \quad (50)$$

where $\mathbf{I}$ and $\mathbf{R}$ are the infection and recovery times, respectively, $n_I$ and $n_R$ are the total number of individuals in the model that are infected and that recover over the course of the simulation, respectively, and $\phi_1$ is the time of the first infection.

To perform S-ABC, we bring all three channels of the multivariate stream — the number of infected individuals, number of recovered individuals, and time — into the range $[0, 1]$ by dividing by $Z$, $Z$, and $T$, respectively. For W-ABC ("Wasserstein"), we set $\lambda = 2$, since the expected vertical range is approximately twice that of the horizontal range $T = 50$ when $Z = 100$.

## D.7 FURTHER EXPERIMENT: THE BROCK & HOMMES MODEL

In this experiment, we consider a heterogenous agent model proposed by Brock and Hommes [1998] which simulates the dynamics of a set of traders operating under different trading strategies. The system of coupled equations comprising the model may be written succinctly with the following transition density:

$$p_{\boldsymbol{\theta}}(\mathbf{y}_{t+1} \mid \mathbf{y}_{1:t}) = \mathcal{N}\left\{ f(\mathbf{y}_{t-2:t}, \boldsymbol{\theta}), \frac{\sigma^2}{R^2} \right\},$$

where

$$f(\mathbf{y}_{t-2:t}, \boldsymbol{\theta}) = \frac{1}{R} \sum_{j=1}^J \frac{\exp\left\{ \beta (\mathbf{y}_t - R\mathbf{y}_{t-1})(g_j \mathbf{y}_{t-2} + b_j - R\mathbf{y}_{t-1}) \right\}}{\sum_{j'=1}^J \exp\left\{ \beta (\mathbf{y}_t - R\mathbf{y}_{t-1})(g_{j'} \mathbf{y}_{t-2} + b_{j'} - R\mathbf{y}_{t-1}) \right\}} (g_j \mathbf{y}_t + b_j)$$

and $R, \beta, \sigma$ are parameters. In this way, we are able to obtain an approximate ground truth posterior with standard Markov chain Monte Carlo (MCMC) techniques such as MH. We follow Platt [2020], Dyer et al. [2024] and assume the following parameter values: $J = 4, R = 1.0, \sigma = 0.04, \beta = 10, g_1 = b_1 = b_4 = 0$ and $g_4 = 1.01$.

The parameters $g_j \in \mathbb{R}$ capture the trend-following tendencies of the agents, while the parameters $b_j \in \mathbb{R}$ determine the biases towards different trading strategies. In our experiments, we consider the task of estimating the posterior $\pi(\boldsymbol{\theta} \mid \mathbf{y})$, where $\boldsymbol{\theta} = (g_2, b_2, g_3, b_3)$, $\mathbf{y} = (\mathbf{y}_1, \dots, \mathbf{y}_n) \sim p_{\boldsymbol{\theta}^*}$ is the pseudo-observation, $n = 100$, and $\boldsymbol{\theta}^* := (-0.7, -0.4, 0.5, 0.3)$ is the parameter setting used to generate $\mathbf{y}$.

We show in Figure 6 boxplots for the Wasserstein distance and MMD between the ABC posteriors, denoted with $\hat{\pi}_{\mathrm{ABC}}$, and the approximate ground-truth posterior obtained with MH, denoted with $\hat{\pi}_{\cdot\mid\mathbf{y}}$. We also show boxplots for the Euclidean distance between the ABC posterior means and the MH posterior mean. These boxplots were created by running the rejection ABC (REJ-ABC) algorithm with the same 20 random seeds. In this experiment, SA-ABC uses the first and second powers of $l$ evenly spaced order statistics of the output data $\mathbf{x}$, as considered in Fearnhead and Prangle [2012], where we take $l = 10$.

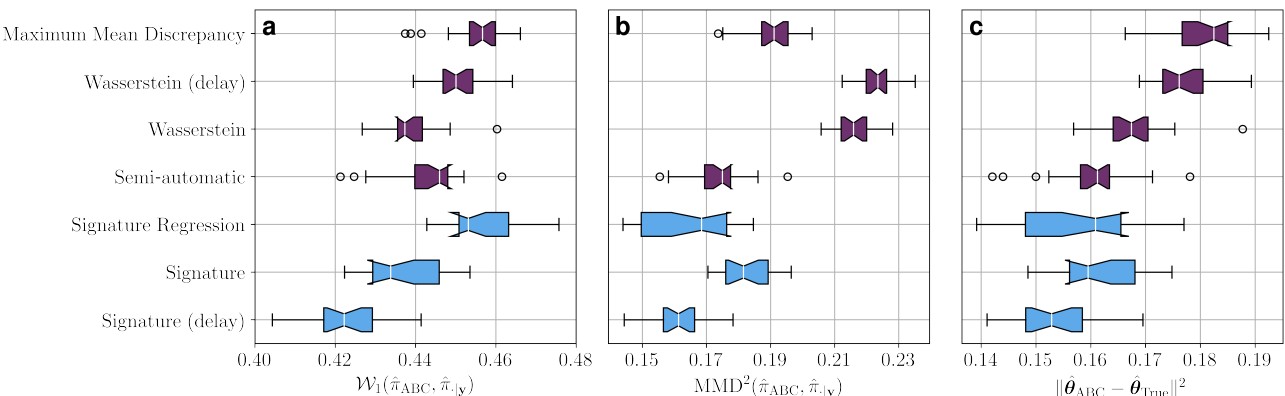

Figure 6: (Brock & Hommes) (**a**) Wasserstein distances between the posteriors recovered from the different distance measures and samples from the exact posterior. (**b**) Maximum mean discrepancies between the posteriors recovered from the different distance measures and samples from the exact posterior. (**c**) Squared distances between the means of the ABC posteriors and the exact posterior mean. Our methods are shown in blue.

From this, we see that the signature-based methods tend to generate lower values in all three metrics compared to existing methods. In particular, we see that S-ABC with the lag-1 delay transformation once again dominates existing methods uniformly across all three metrics, while the same transformation applied to Wasserstein distance (Wass) does not result in the same improvement. This demonstrates the potential power of our signature-based methods as automatic distance measures for ABC for dynamic, stochastic simulators.