# OpenReview forum: "Approximate Bayesian Computation with Path Signatures"
_auai.org/UAI/2024/Conference — UAI 2024 spotlight_

### Official Review · Reviewer_Miym · 2024-03-18

**Q2-1 Originality-Novelty:** 2
**Q2-2 Correctness-Technical Quality:** 3
**Q2-5 Clarity Of Writing:** 4

**Q1 Summary And Contributions:**

Motivation: ABC is a framework for the likelihood-free Bayesian inference of posterior distributions, often used for the inference of simulation parameters. At the core of the framework lies the necessity for defining a distance measure between observations (or summary statistics thereof) and synthetic data. Currently employed distance measures are not made for the express purpose of time series data, hence their performance suffers in these domains.

Contribution: The authors make use of the path signature kernel trick to summarize time-series data for comparison within ABC. Specifically, the signature kernel is used in two ways: First, the signature of a time series is directly used as its summary statistic. Second, the signatures are used to train a regression model for the semi-automatic construction of summary statistics. The authors benchmark these two approaches in comparison to standard distance measures in a number of experiments.

**Q2-3 Extent To Which Claims Are Supported By Evidence:**

3: Good: the main claims are supported by convincing evidence (in the form of adequate experimental evaluation, proofs, (pseudo-)code, references, assumptions).

**Q2-4 Reproducibility:**

2: Fair: key resources (e.g. proofs, code, data) are unavailable but key details (e.g. proof sketches, experimental setup) are sufficiently well-described for an expert to confidently reproduce the main results.

**Q3 Main Strengths:**

ABC is the current simulation-based inference method of choice in many fields, with very active research around ways to improve its sample efficiency. Hence, novel approaches as presented here have potential to significantly impact real-world applications if they can be shown to work well. The authors have provided a lot of evidence for the soundness of the method as well as experimental evidence of its benefit. The approach is well motivated and carefully backed with theoretical proofs as well as experimental evidence.

**Q4 Main Weakness:**

While the authors state that the signature kernel currently incurs higher computational cost than alternative methods, a comparison is not provided. This makes the practicality of the method hard to judge. The experimental evidence is not fully convincing in the approach significantly outperforming other approaches, as the results often seem quite close depending on the metric.

**Q5 Detailed Comments To The Authors:**

This is a well-written paper with strong theoretical backing that introduces a novel way of summarizing time-series data for ABC methods. My question to the authors: How much more computationally costly is this approach currently in comparison? Further, I think it might be useful to walk the reader through an example of a signature kernel in the supplements of the paper. Finally, I recommend to provide the code of the experiments as a supplement if at all possible, to improve reproducibility.

**Q9 Complying With Reviewing Instructions:**

Yes

---

> ### Author Rebuttal · Authors · 2024-04-04
>
> Thank you for reviewing our paper and for your helpful feedback.
>
> **Cost**
>
> Regarding your comment
>
> > While the authors state that the signature kernel currently incurs higher computational cost than alternative methods, a comparison is not provided
>
> and question
>
> > How much more computationally costly is this approach currently in comparison?
>
> Our revised article will include a detailed breakdown of the costs we observed with each method in a table. To give an indication here: in the experiments in Section 4, the average time associated with the S-ABC approach ranged from being 4x _faster_ than the Wasserstein ABC approach in the GBM experiment, to 50x slower than the Wasserstein ABC approach in the GSE model. In comparison to K2-ABC, S-ABC ranged from approximately 10x _faster_ in the GSE experiment to 5x slower than K2-ABC in the Ricker model experiment. Thus S-ABC was faster than Wasserstein ABC and K2-ABC in some experiments but slower in others. SR-ABC, on the other hand, was consistently more costly than other methods, being 10x more expensive than Wasserstein ABC and K2-ABC for the GBM model and approx. 500x more expensive in the Ricker model. This is due to
>
> 1. the need to compute and invert the Gram matrix during training of the kernel ridge regressor, and
> 2. the requirement to compute the signature kernel between each simulated time series and all $R$ training points used to train the kernel ridge regression model (see Equation 16).
>
> These problems are common for kernel methods, and there are a number of good techniques available for reducing cost (e.g., the Nyström method or random Fourier features – see, e.g., Williams and Seeger, 2000; Rahimi and Recht, 2007; Yang et al., 2012) in practical implementations. We will use some of the additional 2 pages made available to us to discuss these options for reducing the cost of SR-ABC in practical implementations.
>
> (It is also worth pointing out that the difference in cost is also likely to be partly, if not largely, down to implementation: our code uses the first ever release of the [`sigkernel` package](https://github.com/crispitagorico/sigkernel) for the signature kernel computations, which to the best of our knowledge was the first ever open-source implementation of the signature kernel. In contrast, the Wasserstein distance computations are performed using the well-developed and mature [Python Optimal Transport package](https://github.com/PythonOT/POT). Thus it is likely that this observed difference in cost will reduce, or may even vanish in practical terms, as more efficient implementations of signature kernel computations emerge and are used in future implementations of S-ABC and SR-ABC.)
>
> **Intuition for and examples of signatures**
>
> Regarding your suggestion that
>
> > it might be useful to walk the reader through an example of a signature kernel in the supplements of the paper
>
> We agree: we will use some of the additional 2 pages allowed to move Example 1 and Figure 4 from Appendix B.1 into Section 2.2, to improve accessibility and provide intuition about the information captured by the path signature. We will also provide further details on how the signature kernel is computed in the supplement, as suggested.
>
> **Release of code**
>
> Regarding your comment
>
> > Finally, I recommend to provide the code of the experiments as a supplement if at all possible, to improve reproducibility
>
> We agree: code for all experiments is available in anonymised form at https://anonymous.4open.science/r/s-abc-anon-75DD and will be released with the paper on GitHub upon acceptance to improve reproducibility.
>
> Thank you once again for having taken the time to provide us with helpful feedback.
>
> **References**
> 1. Williams, Christopher, and Matthias Seeger. "Using the Nyström method to speed up kernel machines." Advances in neural information processing systems 13 (2000).
> 2. Rahimi, Ali, and Benjamin Recht. "Random features for large-scale kernel machines." Advances in neural information processing systems 20 (2007).
> 3. Yang, Tianbao, et al. "Nyström method vs random fourier features: A theoretical and empirical comparison." Advances in neural information processing systems 25 (2012).

---

### Official Review · Reviewer_ytxr · 2024-03-20

**Q2-1 Originality-Novelty:** 3
**Q2-2 Correctness-Technical Quality:** 3
**Q2-5 Clarity Of Writing:** 3

**Q1 Summary And Contributions:**

In appproximate Bayesian computations (ABC) applied to time series data, it may be difficult to measure the distance between actual samples and simulator model output, which is necessary for ABC. This is particularly so because of the non-iid nature of time series data. The paper uses path signatures, a representation of multi-variate and irregularly sampled time series data into a unique summary or signature of the time series through Euclidean space. Two approaches to incorporating path signatures in ABC are introduced: to use it directly as a summary statistics or to use it for the regressing task in ABC. Experiments show better performance compared to existing distance measures.

**Q2-3 Extent To Which Claims Are Supported By Evidence:**

3: Good: the main claims are supported by convincing evidence (in the form of adequate experimental evaluation, proofs, (pseudo-)code, references, assumptions).

**Q2-4 Reproducibility:**

3: Good: key resources (e.g. proofs, code, data) are available and key details (e.g. proofs, experimental setup) are sufficiently well-described for competent researchers to confidently reproduce the main results.

**Q3 Main Strengths:**

The paper is in general well written and understandable for an audience knowledgeable with ABC for time series data; the research question follows from the challenges in the ABC approach for non-iid sequential data. The presentation is in general clear although requires substantial background knowledge from the reader to understand the details of the proofs, experiments, and methods. The experimental results, although limited, show a clear improvement over other methods.

**Q4 Main Weakness:**

Although I consider this a solid contribution there are a lot of technical details that withhold a reader not familiar with them to grasp the main contributions and novelty of the approach; for example, a brief non-technical intuitive introduction of path signatures may help the non-expert in this particular sub-field. The implementation of the new methods leads to higher computational costs in practice than comparable methods, so clearly work here is needed to improve efficiency of the implementation.

**Q5 Detailed Comments To The Authors:**

I would suggest to try to give a bit more intuition behind the key concepts you use to make the paper accessible for a broader audience. Minor points: p2 before section 2: in this paragraph it is not intuitively clear that both approaches use path signatures in different ways. you may want to rephrase that to make this more clear. Pag 3 just before 2.2: "sill" might be a typo. p8 conclusion line 6: "and that" seems to refer to object, not method.

**Q9 Complying With Reviewing Instructions:**

Yes

---

> ### Author Rebuttal · Authors · 2024-04-03
>
> Thank you for your kind words and for your helpful review.
>
> Regarding your comment that
>
> > a brief non-technical intuitive introduction of path signatures may help the non-expert in this particular sub-field
>
> and suggestion to
>
> > try to give a bit more intuition behind the key concepts you use to make the paper accessible for a broader audience
>
> We agree: we will use some of the additional 2 pages to expand on the introduction to path signatures in Section 2.2 with examples to provide the reader with better intuition. We will achieve this by bringing Example 1 and Figure 4 from Appendix B.1 into the main text and incorporating them into Section 2.2, before Section 2.2.1.
>
> Regarding your comment that
>
> > the new methods leads to higher computational costs in practice than comparable methods, so clearly work here is needed to improve efficiency of the implementation
>
> we will use some of the additional space to provide a more extended discussion of this point than was possible in the original submission due to space constraints. For example, we will discuss how standard techniques from kernel methods such as the Nyström approximation and random Fourier features can be used to reduce the cost of the SR-ABC method (which is the main source of increased cost relative to the baselines) in practical implementations of the method.
>
> Regarding your detailed comments:
>
> > p2 before section 2: in this paragraph it is not intuitively clear that both approaches use path signatures in different ways. you may want to rephrase that to make this more clear.
>
> Thank you – we will rephrase this sentence to clarify that we propose to use path signatures in two different ways for ABC.
>
> > Page 3 just before 2.2: "sill" might be a typo.
>
> Thank you for bringing this typo to our attention – this should indeed say “still” rather than "sill". We will correct this in the revised paper.
>
> > p8 conclusion line 6: "and that" seems to refer to object, not method.
>
> Thank you for pointing this out too – we will fix this to say “which” rather than “and that”.
>
> Thank you once again for having taken the time to provide helpful feedback on our work.

---

### Official Review · Reviewer_m543 · 2024-03-22

**Q2-1 Originality-Novelty:** 3
**Q2-2 Correctness-Technical Quality:** 3
**Q2-5 Clarity Of Writing:** 3

**Q1 Summary And Contributions:**

The paper studies the approximate estimation of the posterior distribution in Bayesian inference by applying the path signature [e.g., Lyons et al., 2007]. It proposes two methods: the signature ABC (S-ABC) and the signature regression ABC (SR-ABC), and shows their superiocy mainly through case studies. The S-ABC method leverages the fact that the path signature is a sufficient statistic and uses it as a summary statistic. The SR-ABC, on the other hand, utilizes the training examples to estimate the summary statistic.

**Q2-3 Extent To Which Claims Are Supported By Evidence:**

4: Excellent: all claims are supported by very convincing evidence (in the form of comprehensive experimental evaluation, rigorous mathematical proofs, detailed (pseudo-)code, precise references, well-motivated and realistic assumptions) and the authors deliver what they promise.

**Q2-4 Reproducibility:**

4: Excellent: key resources (e.g. proofs, code, data) are available and key details (e.g. proof sketches, experimental setup) are comprehensively described for competent researchers to confidently and easily reproduce the main results.

**Q3 Main Strengths:**

- In general, the paper is well structured and clearly written.
- The paper summarizes the known methods for Approximate Bayesian Computation (ABC) in the background section.
- An extended background on path signatures is included in the appendix. I found the example there quite helpful.
- The soundness of S-ABC is stated as a proposition and is proved in the Appendix.

**Q4 Main Weakness:**

- From my understanding, most methods and frameworks such as path signatures, rejection ABC, and semi-automatic ABC were known. Hence, the main novelty here is to adopt the concept of path signature to the existing approaches of ABC.
- The definition of path signatures is kind of abstract, so examples will be very helpful. Maybe move Example 1 from Appendix to the main paper.
- see detailed questions below.

**Q5 Detailed Comments To The Authors:**

1. In Section 2 Background paragraph of Semi-automatic ABC, "A drawback ... the construction of an initial set of candidate summaries ..." I found it difficult to interpret this sentence. Are "candidate summaries" referring to the training examples?

2. In Section 2.2, more explanations on "H-valued paths" will be helpful. Also, I don't think {t1,...,tn} should be called a "partition" since they are not sub-intervals.

3. Equation (16). If I understood correctly, the complexity of computing Equation (16) $\theta_j$ grows with the number of training examples? If so, computing the inverse of RxR matrix (for $\omega^{(j)}$) can be very expensive here. Would this affect the scalability of the algorithm? This may be a good add-on to Section 4.5.

4. Any intuition on why SR-ABC did worse than S-ABC in Figure 2?

Typos:
- page 2, paragraph on K2-ABC, "the choice of summary statistics ... replaced by the choice of kernel $k$", $k$ -> $\kappa$.

**Q9 Complying With Reviewing Instructions:**

Yes

---

> ### Author Rebuttal · Authors · 2024-04-04
>
> Thank you for your helpful comments and for reviewing our paper.
>
> Regarding your comment that
>
> > The definition of path signatures is kind of abstract, so examples will be very helpful. Maybe move Example 1 from Appendix to the main paper
>
> Thank you for the suggestion; we agree entirely, and will use some of the extra 2 pages available in the revision to move Example 1 and Figure 4 into Section 2.2, to provide the reader with more intuition and a clear example of the kind of information the signature extracts from paths.
>
>
> Regarding your detailed comments:
>
> 1. We will also use some of the additional space in the revision to clarify this point. (Briefly: Our point here is that semi-automatic ABC allows the experimenter to learn a low-dimensional vector $\mathbf{s}$ of summary statistics through a regression task, in which the $\boldsymbol{\theta}^{(i)}$ are regressed onto $\mathbf{g}(\mathbf{x}^{(i)})$, where $\mathbf{g}(\mathbf{x}^{(i)})$ is a vector of "candidate" summary statistics of $\mathbf{x}^{(i)}$. However, this still requires the experimenter to devise such a $\mathbf{g}$ themselves to use as regressors, and it can be hard for the experimenter to know what a good choice of $\mathbf{g}$ is. This motivates the use of path signatures to perform this regression task: path signatures provide a ready-made set of regressors – obviating the need for the experimenter to come up with any choice of $\mathbf{g}$ themselves – that have good function approximation abilities due to their universal nonlinearity property discussed in Section 2.2.)
> 2. We will provide more explanation on what is meant by $\mathcal{H}$-valued paths. The example we will bring from Appendix B.1, as per your suggestion, will help to demonstrate the case where $\mathcal{H} = \mathbb{R}^d$. Regarding your point about terming $\lbrace{t_1, \ldots, t_n\rbrace}$ a “partition”: while we do understand your point, “partition” is standard terminology for this in the signature kernel literature (see, e.g., Kiraly and Oberhauser, 2019; Salvi et al., 2021) and so we would rather keep this terminology. However, we would be willing to consider alternative terms (e.g., “sequence”?) if you feel strongly about this.
> 3. It is correct that the complexity of computing Equation 16 grows with the number of training examples, and that this does affect the scalability of the method (such problems are common for kernel methods). We will use some of the additional 2 pages to provide a discussion of this in Section 4.5, along with
>     1. a more detailed discussion of how this led to an increased computational cost of our SR-ABC approach compared to the other methods we consider, and
>     2. how this increased cost can be alleviated using well-known cost-reduction techniques from the kernel literature (e.g., the Nyström method or random Fourier features – see, e.g., Williams and Seeger, 2000; Rahimi and Recht, 2007; Yang et al., 2012) that we did not specifically explore in our implementations.
> 4. I’m afraid we do not have a strong intuition about why this is the case. It is possible that this is just an inference task that semi-automatic approaches have difficulty with – notice that “Semi-automatic” also performs poorly in comparison to the W-ABC methods in this example.
>
> Finally, thank you for pointing out the typo, which should indeed be $\kappa$ rather than $k$ in the description of K2-ABC on page 2. We will correct this.
>
> Thank you once again for your helpful comments and for having taken the time to review our paper.
>
> **References**
> 1. Király, Franz J., and Harald Oberhauser. "Kernels for sequentially ordered data." Journal of Machine Learning Research 20.31 (2019): 1-45.
> 2. Salvi, Cristopher, et al. "The Signature Kernel is the solution of a Goursat PDE." SIAM Journal on Mathematics of Data Science 3.3 (2021): 873-899.
> 3. Williams, Christopher, and Matthias Seeger. "Using the Nyström method to speed up kernel machines." Advances in neural information processing systems 13 (2000).
> 4. Rahimi, Ali, and Benjamin Recht. "Random features for large-scale kernel machines." Advances in neural information processing systems 20 (2007).
> 5. Yang, Tianbao, et al. "Nyström method vs random fourier features: A theoretical and empirical comparison." Advances in neural information processing systems 25 (2012).

---

### Official Review · Reviewer_wTBH · 2024-03-23

**Q2-1 Originality-Novelty:** 3
**Q2-2 Correctness-Technical Quality:** 3
**Q2-5 Clarity Of Writing:** 3

**Q10 Ethical Concerns:**

No.

**Q1 Summary And Contributions:**

The paper proposes two procedures for approximating the likelihood function of simulator models using a kernel function based on path signatures, a set of statistics capturing geometric features of functions over time. The first procedure incorporates a signature kernel directly in rejection-based Approximate Bayesian Computation (ABC). The second procedure regresses system paths onto system parameters using signature kernel regression, and uses the regression function as a summary statistic in ABC. Both procedures ultimately produce approximate posterior distributions over parameters of stochastic dynamical systems. These are compared to popular alternative ABC methods on three different systems and results indicate strong performance.

**Q2-3 Extent To Which Claims Are Supported By Evidence:**

4: Excellent: all claims are supported by very convincing evidence (in the form of comprehensive experimental evaluation, rigorous mathematical proofs, detailed (pseudo-)code, precise references, well-motivated and realistic assumptions) and the authors deliver what they promise.

**Q2-4 Reproducibility:**

3: Good: key resources (e.g. proofs, code, data) are available and key details (e.g. proofs, experimental setup) are sufficiently well-described for competent researchers to confidently reproduce the main results.

**Q3 Main Strengths:**

- The proposed procedure is a very general solution to the problem (i.e., any continuous real-valued function over time with bounded variation can be approximated with signatures and signatures are a sufficient statistic, thus naturally fitting the general ABC formulation).
- The proposed procedure is clearly positioned within the literature.
- Experimental results indicate strong performance.

**Q4 Main Weakness:**

- The path signature description section is rather abstract; examples of "geometric features of paths" or illustrations would have been helpful.
- A computational description of the signature kernel is missing.

**Q5 Detailed Comments To The Authors:**

- (Eq. 3) Shouldn't $\epsilon$ be in the likelihood subscript as well?
- (Proposition 1) In the first equation, the supremum over $\theta$ of $p_{\theta}(x)$ is still a function of $x$. Should it therefore not be that $\sup_{\theta} p_{\theta}(x) < \infty$ for all $x$?
- (Sec. 4.2) The selected Uniform priors are rather narrowly contained around the true parameters. How does the procedure perform with non-informative priors?

**Q9 Complying With Reviewing Instructions:**

Yes

---

> ### Author Rebuttal · Authors · 2024-04-03
>
> Thank you for reviewing our paper and for your helpful feedback.
>
> To address your comment that
>
> > The path signature description section is rather abstract; examples of "geometric features of paths" or illustrations would have been helpful.
>
> we will use part of the additional 2 pages to move Example 1 and Figure 4 from Appendix B.1 into Section 2.2, to provide an example and improve intuition about the information captured by the signature from paths.
>
> We will also provide further details (most likely in an appendix, if not in Section 2.2.1) on how the signature kernel is computed in practice to address your comment that
>
> > A computational description of the signature kernel is missing.
>
> Regarding your detailed comments:
>
> - The likelihood function appearing in Equation 3 should indeed also have a $\varepsilon$ in the subscript. Thank you for pointing out this typo.
> - This is also a typo: rather than $\mathbf{x}$, the observed data $\mathbf{y}$ (which is fixed) is what should appear in this expression. Thank you for pointing out this typo too.
> - Regarding your question “How does the procedure perform with non-informative priors?”: the example presented in Section 4.4 uses relatively uninformative Gamma priors (in the sense that its tails span multiple orders of magnitude beyond the value of the “true parameters”) and we see that the method still performs well in this case. This provides us with confidence that the method will work well in other cases with less informative priors.
>
> Thank you once again for your help in improving our submission.

---

### Official Review · Reviewer_7wRj · 2024-03-24

**Q2-1 Originality-Novelty:** 3
**Q2-2 Correctness-Technical Quality:** 3
**Q2-5 Clarity Of Writing:** 2

**Q1 Summary And Contributions:**

Stochastic simulation models used throughout many fields typically lack a tractable likelihood, requiring approximative methods such as Approximate Bayesian Computation (ABC) to infer model parameters based on data. A well-known problem with ABC is that it requires carefully crafted summary statistics to determine whether a model output is "close" to the provided data, as ABC performance strongly depends on how informative these summary statistics are. Recent methods have attempted to circumvent the need for hand-crafted summary statistics, but these typically assume iid series and as such, do not work well for time series. This makes these methods of limited use for simulation models with dynamic outputs, which are widespread in many scientific disciplines.

The paper proposes the use of path signatures as a universal summary statistic for time series data. The authors present two variations of ABC that each rely on path signatures to infer model parameters based on dynamic data. This is an interesting approach to an open problem that could be of value to a large scientific community if successful.

**Q2-3 Extent To Which Claims Are Supported By Evidence:**

3: Good: the main claims are supported by convincing evidence (in the form of adequate experimental evaluation, proofs, (pseudo-)code, references, assumptions).

**Q2-4 Reproducibility:**

2: Fair: key resources (e.g. proofs, code, data) are unavailable but key details (e.g. proof sketches, experimental setup) are sufficiently well-described for an expert to confidently reproduce the main results.

**Q3 Main Strengths:**

**Impact** : improving ABC performance for dynamic models without hand-crafted summary statistics can have a large impact in many fields that use (stochastic) simulation models to reason about dynamic processes. The approach taken here seems interesting and could potentially solve this issue (although I am not yet sure how generically applicable this method is based on the examples presented here).

**Q4 Main Weakness:**

**Reproducibility**

No code is provided. The authors mention that the distance between path signatures "can be computed easily through existing software", but since path signatures are not (yet) widespread in their use, I still think the paper would be much more usable if example code could be provided. This would be very helpful for other scientists that want to try this approach on their simulations.

**Support of claims by evidence**

One of the main claims of the paper is that path signatures are a sufficient statistic for time series data,  which is what makes them attractive for use with ABC. But this seems to contradict the fact that path signatures are invariant to time transformations. For some (or even many?) dynamic model outputs the speed at which things change might in itself be an important signal that could inform model parameters, but this would not be picked up by the (default) path signature (unless I misunderstood?). The same could be said for the translation invariance; there might be cases where starting point matters. The authors state in the appendix that there is a way around this, but since the paper presents the path signature as a universal statistic to be used with ABC, I think it would be important to discuss in the main paper when edits are in order.

**Sometimes more context needed**

- Why were these specific example models chosen? Which features make them interesting/representative? It would have been nice to see some examples of the time series themselves rather than only the posterior estimation metrics displayed in the paper, to get an idea of how similar/dissimilar model dynamics are. Are there other dynamic models where this would not work or would be harder? Were these easy or hard models to fit?

- Since results are stochastic, it would be informative to also show the distance metric for different simulations under the "correct" parameters $\theta$ as a lower bound. Even if the parameters are recovered perfectly, we still will not find a distance of zero, so this would provide some idea of how close we are to the best-possible solution.

- What is the idea behind the "delay" approach? This is shown in several figures but not explained in the (main) paper.

- Which signature-based approach should be used when? The paper introduces the two approaches but does not discuss when to use what. In the experiments, signature regression seems more variable between runs, why is this?

**Difficult to read**

This might be my lack of background in tensor algebra, but I found some parts of the paper quite difficult to read. If the other reviewers disagree I will slightly raise my score, but if possible I think it would be beneficial to add some intuition here and there for scientists who use simulation models and might want to use this method, but are (like me) not experts in tensor algebra. This would improve accessibility of the paper. There also seem to be some minor inconsistencies; see Q5 for details.

**Q5 Detailed Comments To The Authors:**

Most important questions:

- At some points the paper discusses "signature feature map" - what is meant by this, is this just the signature itself?

- The paper sometimes uses $k$ to denote a kernel and sometimes $\kappa$ - what is the difference? Is one for the whole path $h$ and one for the observations $x$? Or is there some other difference? And related: this might be a stupid question but what do you mean with the dot in e.g. $\kappa( x_t, \cdot)$?

- Section 2.2.1 states that we can apply the kernel to observation points only. But there must be some interpolation going on, and therefore an approximation of the signature of the true path? I don't understand how you can then guarantee that you approximate the true signature well. For example, in the extreme you could have a curved path with only two observation points, in which case there is no way to retrieve the geometry (and thus signature) of the true underlying path unless you can make strong assumptions. So how strongly does sampling rate affect how well the "observation-based" path signature approximates the "true path signature" of the underlying path? Does this decrease or increase at higher signature "levels" or is there no such trend? Is this important to take into account in ABC?

- Section 4.4: Why now a different presentation of the outcomes compared to the other models?


Minor questions/comments:

- Section 2.2 introduces $\zeta(0,T) =  t1, ... , tn $, a finite partition of time points in the interval [0,T]. But how is this different from $\cal{X}$ introduced in section 2.2.2? If these are not the discrete points where data $x = (x_{t1}, ..., x_{tn})$ are sampled, then how should I interpret $\zeta$?

- Eqn 7: $m$ is used/introduced here but only explained a few paragraphs later, this should be explained here.

- Why are distances sometimes denoted $\rho$ and sometimes $\cal{D}$? Is there a difference?

- What is $\delta$ in "Results" in algorithm 1?

- MH - I assume this is Metropolis-Hastings, but this should perhaps be explained here (unless I missed it, but I don't think the abbreviation was introduced before).

-  Section 4.4: population size is denoted both $N$ and $Z$. Is this an error or do they represent something different?

- Section 4.4: This might be me but I don't understand the model from this notation - could you explain further?

- $\Gamma$ distributions: which parametrization (shape-scale or shape-rate)?

- "pooling the best $M$ distances" in Fig 3 - so what $M$ was used?

**Q9 Complying With Reviewing Instructions:**

Yes

---

> ### Author Rebuttal · Authors · 2024-04-04
>
> Thank you for your helpful review. Brevity $\neq$ curtness: 5k characters only!
>
> **Reproducibility**
>
> Code goes on GitHub if accepted: https://anonymous.4open.science/r/s-abc-anon-75DD
>
> **Support of claims**
>
> We'll discuss this in the main text using the extra space. Our results (e.g., Appendix C) assume time- & basepoint-augmentation (we'll define these in main text too) to remove invariances; these can always be used so no issues
>
> **Context**
> * We'll discuss these points. E.g., we chose our examples for
>   1. variety of application domains (ecology, finance, social/epidemiological)
>   2. availability of good approximate ground-truth posteriors, permitting proper assessment of the ability to recover the _full_ target posteriors (rather than parameter point estimates alone)
>   3. variety of model outputs (Ricker: chaotic, integer-valued series; GBM: non-stationary, non-ergodic, continuous output; Section 4.4: continuous-time, irregularly spaced points) to demonstrate signature’s versatility & compatibility with messy real-world settings
> * Our understanding of your suggestion is to:
>   1. compute distances between real time series $\mathbf{y}$ & samples from posterior predictive distributions
>   2. compare to distances achieved by samples generated from a point mass on the true parameters
>
>   We don't do this because this doesn't necessarily indicate good performance: true posteriors can be diffuse, in which case it may be _correct_ for the posterior predictive to produce samples with large distances from $\mathbf{y}$
> * "Delay" defined in Appendix B.5; we'll use extra space to move to main text
> * We'll use extra space to discuss pros/cons of S-ABC & SR-ABC in Conclusion, e.g.:
>   - SR-ABC trains a regression model; S-ABC doesn't. So latter generally faster & former introduces more stochasticity (why SR-ABC is more variable, we think)
>   - Semi-automatic methods (e.g., SR-ABC) aim to recover point estimates well (e.g., posterior mean; [Fearnhead and Prangle, 2012](https://rss.onlinelibrary.wiley.com/doi/10.1111/j.1467-9868.2011.01010.x))); other ABC methods (e.g., S-ABC) aim to recover the full posterior well. So the choice depends partly on what the experimenter is interested in
>
> **Difficult to read**
>
> We'll use extra space to move Example 1 & Fig. 4 from Appendix B.1 to Section 2.2 to improve accessibility
>
> **Detailed Comments**
>
> Most important:
> * "Signature feature map" = "signature". We'll edit to just use “signature”
> * We'll use extra space to clarify:
>   - $\mathcal{X}$: the space that data points $\mathbf{y}_t$ & $\mathbf{x}_t$ comprising time series $\mathbf{y}$ & $\mathbf{x}$ take values in
>   - $\kappa$: inner product on $\mathcal{X}$
>   - $k$: signature kernel; inner product between signatures of paths/_sequences_ of points interpolated to form a path. We explain in Section 2.2.1 that $k$ “sequentialises” $\kappa$, meaning $k(\mathbf{x}, \mathbf{y})$ can be evaluated only by evaluating $\kappa(\mathbf{x}_i, \mathbf{y}_j)$ for different pairs of points $\mathbf{x}_i$ & $\mathbf{y}_j$ in $\mathbf{y}$ & $\mathbf{x}$
>   - Dot in $κ(\mathbf{x}_t, \cdot)$ is a free argument. $κ(\mathbf{x}_t,\cdot)$ represents the point in $\kappa$'s RKHS $\mathcal{H}$ to which $\mathbf{x}_t$ is mapped
>
> * We use linear interpolation (see Section 2.2.2 & Eq. 13). This is standard & simplifies signature kernel computations; see Thm 2 [Kiraly and Oberhauser (2019)](https://www.jmlr.org/papers/v20/16-314.html). We don't guarantee good approximation of signatures of underlying paths; we only guarantee injective map from (i.e., a sufficient statistic of) the _observed_ points in $\mathbf{x}$ & $\mathbf{y}$. Success here requires only that all information about the _observed_ points is kept, not that signatures of any true underlying paths are reconstructed well. (For many models there is not even necessarily a fixed notion of a true underlying path, e.g., the discrete event/discrete time simulators in Sections 4.2 & 4.4, so no fixed meaning to "true signature")
>
> * Format change here because values for W-ABC & S-ABC are far smaller than those for K2-ABC. Table is clear but boxplots would be illegible
>
> Minor:
> - $\zeta(0,T)$: an _arbitrary_ finite partition of interval $[0,T]$, not necessarily one corresponding to any $\mathbf{x}$. $\mathcal{X}$ explained above
> - $m$: an arbitrary index. We’ll define “depth” closer to Eq. 7
> - We use $\rho$ (resp. $\mathcal{D}$) to denote a distance on the space of signatures (resp. on $\mathcal{X}^n$). We'll clarify this & check for consistency. (To be fair, Eq. 14 & Prop. 1 are perhaps better written with $\mathcal{D}$ to be consistent with this rule)
> - We'll define $\delta$ = Dirac mass on its subscript
> - We'll define “MH” = “Metropolis-Hastings”
> - Typo: $Z$ is population size, $N$ is no. of samples in Rejection ABC
> - We’ll use extra space to give alternative formulation
> - Shape-rate. We'll clarify
> - $M = 100$ used in each of 20 runs (stated on p.7). To avoid ambiguity we'll add “totalling $2000$ samples”

---

### Meta-Review · Area_Chair_WFQo · 2024-04-16

A clear and solid consensus by the reviewers recommending accept for a paper that proposes an interesting approach to simulation-based inference/ABC. Please update the paper with the promised changes when revising the paper.